# Self /other recognition and distinction in adolescents with anorexia nervosa: A pilot study using a double mirror paradigm

Nathalie Lavenne-Collot[1,2,3]*, Emilie Maubant[1,2], Stéphanie Déroulez[1],
Guillaume Bronsard[1,2,4], Moritz Wehrmann[5,6], Michel Botbol[2,7], Alain Berthoz[8,9]

1 Département de Psychiatrie de l'Enfant et de l'Adolescent, CHRU Brest, Brest, France, 2 Université de
Bretagne Occidentale, Brest, France, 3 Laboratoire du traitement de l'information Médicale, Inserm U1101,
Brest, France, 4 Département de Sciences Humaines et Sociales, EA 7479, EA 3279 (CEReSS, AMU),
Brest, France, 5 Laboratoire de Physiologie de la Perception et de l'Action UMR 7152 CNRS, Collège de
France, Paris, France, 6 Bauhaus-Universität Weimar, Weimar, Germany, 7 Professeur Emérite de
Psychiatrie de l'Enfant et de l'Adolescent, Université de Bretagne Occidentale, Brest, France, 8 Professeur
Honoraire au Collège de France, Paris, France, 9 Centre Interdisciplinaire de Biologie (CIRB), Paris, France

* nathalie.lavenne-collot@chu-brest.fr

pone.0309548

Illes Balears, SPAIN

**Data Availability Statement:** All relevant data are
within the manuscript and its Supporting
Information files.

## Abstract

### Background

Abnormalities in body perception in patients affected by anorexia nervosa have been widely
studied, but without explicit reference to their relationship to others and the social processes
involved. Yet, there are a several arguments supporting impairments in interpersonal rela-
tionships in these patients. Notably, some evidence suggests that self/other distinction
(SOD), the ability to distinguish one's own body, actions and mental representations from
those of others could be impaired. But research remains scarce in this area.

### Material and methods

A single-centre, prospective pilot study was conducted to investigate, for the first time, self-
recognition and SOD in seven adolescents with anorexia nervosa compared with matched
healthy controls (HCs) using the "Alter Ego"TM double mirror paradigm. This innovative
device allows the progressive morphing of one's own face to that of another and vice versa
between two subjects that interact on opposite sides of the device. Two judgement criteria
were used: 1) M1: the threshold at which subjects start to recognize their own face during
other-to-self morphing, and 2) M2: the threshold at which subjects start to recognize the oth-
er's face during self-to-other morphing. In a second part, SOD was reassessed during five
different sensorimotor tasks aimed at increasing body self-consciousness in participants
with anorexia nervosa.

### Results

The results showed that the participants with anorexia nervosa exhibited earlier self-recog-
nition in the other-to-self sequence and delayed other-recognition in the self-to-other
sequence. Furthermore, in contrast with that of HCs, the critical threshold for switching

**Funding:** The authors received no specific funding for this work.

**Competing interests:** The authors have declared that no competing interests exist.

between self and other varied with the direction of morphing in anorexia nervosa participants. Finally, when participants with anorexia were seated in a chair with a backrest and footrest strengthening the median axis of their body, the self-recognition threshold (M1) increased significantly, approaching that of controls.

## Conclusions

Although additional research is needed to replicate the results of this pilot study, it revealed the first behavioural evidence of altered SOD in individuals affected by anorexia nervosa through an embodied, semiecological face-recognition paradigm. The relationships between anomalies in body perception and alterations in interpersonal relationships are discussed within an integrative framework from phenomenology to neuroscience, and new research and therapeutic perspectives are presented.

## 1 Introduction

Anorexia nervosa (AN) is an increasingly common eating disorder that primarily affects adolescent girls and young women. Abnormalities in body perception and dysmorphophobia are key symptoms [1]. Over the past decade, a growing number of studies have attempted to understand the disturbed experience of the body in AN. Most studies have focused on abnormalities in *body image* [2], a multidimensional construct that encompasses not only perceptual representations of the body but also broader semantic, aesthetic, and emotional aspects [3]. Notably, there is a consensus that patients with AN overestimate their own visually perceived body size (for a review see [4]). However, more recent findings suggest that other levels of body representation, such as *body schema* or *body consciousness*, may also be affected [5]. The body schema, originally conceived by Head and Holmes [6], refers to a dynamic unconscious sensorimotor representation of the body, built on tactile, kinesthetic, visual, and labyrinthine inputs. It is elicited by action, whether imagined, anticipated and/or performed [7]. In the case of AN, distortions in multisensory integration processing that is fundamental to the acquisition and updating of body consciousness [8] have been demonstrated in many sensory modalities (for a review see [9]). Research has indeed shown somato-tactile distortions of height in the horizontal plane [10–12], disturbances in proprioception and kinesthetic processing [13, 14], interoceptive [15] and extraroceptive [16] alterations in sensitivity and awareness, decreased multisensory integration [17–19] as well as deficits in sensorimotor/proprioceptive memory [20].

Despite the great interest of this research, it has mainly focused on the representation of the physical body through evaluations of the body in relation to the environment or even to objects, but without explicit reference to others or to the social processes involved in relations between self and others. Consequently, even recent studies fail to elucidate the links between anomalies in body representation and other work from psychodynamic theories or cognitive sciences showing that anorexia is associated with alterations in socio-cognitive and interpersonal functioning [21]. Notably, theory of mind and emotional empathy may be impaired in anorexia, as well as emotional and social functioning [22–25]. Interestingly, the phenomenological approach to anorexia provides an integrative and multidimensional conception of bodily self-consciousness, which does not leave aside the interpersonal and intersubjective aspects of the disorder that are proeminent in the clinical presentation and relationships with

anorexic [26, 27]. Notably, several authors emphasize an attitude of over-objectification towards the body in patients with anorexia, consisting in adopting the other's perspective from a third-person rather than first-person embodied perspective, as if their body doesn't pertain to their self and a weakening of self-other boundaries [5]. Moreover, even an extensive literature in cognitive neuroscience emphasizes the role of bodily self-consciousness and multisensory integration processing in establishing a clear distinction between self and other [28–31]. Consequently, exploring the relationship between bodily abnormalities and the self/other distinction in anorexia seems to be a promising avenue.

## Self / Other distinction

In recent years, the concept of self/other distinction (SOD) has received growing interest [32–35] and its study is increasingly relevant from a transosographical perspective in psychiatric disorders [36]. Despite a variety of terminology, including "differentiation", "distinction", "switching," or "agency" of self or other, all of the postulated processes seem to share a common characteristic of "controlling" shared representations between self and other [37, 38]. The ability to differentiate one's own body, actions, and mental states from those of others is crucial for establishing relationships with others while maintaining a stable sense of self [8, 35, 39]. Such social experiences involve both an ability to identify with others and an ability to distinguish ourselves from others [40, 41]. In the absence of this capacity for SOD, confusion between self and other can occur: the experience of others can be confused as coming from the self (i.e., "altercentric" bias), or one can assume an understanding of the other's mind based on one's own experience (i.e., "egocentric" bias) [42, 43].

Disruptions in SOD have been previously demonstrated in a number of diseases or conditions such as borderline personality disorder [44], schizophrenia [45], Autism Spectrum Disorders (ASD) [35] and alexithymia [46]. Interestingly, patients with AN show increased levels of alexithymia [47], and there are numerous studies supporting the links between AN and ASD [48].

There are several arguments supporting SOD impairments in individuals with AN. First, pychodynamic works on the pathogenesis of the disorder argues that AN is frequently associated with the notion of "false self" [49], which is a common condition in early childhood in which children perceive their caregivers' thoughts or desires as their own because they cannot distinguish them. Similarly, AN is also frequently associated with a failure in the separation-individuation line of development [50–52]. In addition, findings from cognitive science have shown that AN is associated with an empathy deficit [53], which encompasses difficulty distinguishing between the respective perspectives of self and others and the ability to switch between them [54]. Moreover, authors have highlighted impairments in spatial cognition in AN, including predominant egocentricity during spatial performance tasks [55]. Notably, many studies support the relationship between altered spatial references, perspective changes, and social cognition or empathy [54, 56–58], suggesting egocentricity also in SOD.

Some neurobiological findings further support SOD impairments in AN. In an RMI study, Sachdev explored the neural basis of self-body recognition in AN patients compared to healthy controls. When the two groups were compared in terms of differential activation with self versus non-self images, patients with AN showed no significant region activation with self images relative to baseline [59]. In addition, Vocks et al. exposed AN patients and control subjects to photographs of their own body and to the body of an unfamiliar other person [60]. In the AN group, visualization of their own body was associated with a significant decrease in a complex neural network. According to the authors, these results could reflect a body-related avoidance behavior. However, an alternative hypothesis could be a disturbance of self-recognition since

this neural network is also involved in visual self-recognition [61]. Furthermore, the literature on alterations in social and affective touch in anorexia nervosa is particularly interesting for shedding light on the links between anomalies in body representation and social interactions. Indeed, touch is one of the earliest human experiences that evokes both the perception of the limits of one's own body and a fundamental form of social interaction [62, 63]. Especially, learning to differentiate self from non-self begins reflexively in the womb [64, 65], and tactile cues play a crucial role in early development [66, 67] as well as in social communication and bonding [68]. While affective touch is perceived as most pleasurable in typically developing populations [69], research has shown that the hedonic value of touch is perceived as unpleasant in people with AN [70–72] with neural hypersensitivity persisting even after remission [73]. Furthermore, with regard to the specific issue of self/other distinction, in an fMRI study using a task based on the comparison between self-touch versus other-touch, alterations in neural processing were demonstrated in participants with AN, with tactile sensations produced by the self rendered more similar to those produced by others [74].

To date, studies that have explored SOD generally used prerecorded static images or movies that progressively morph from one's own face to another's face and vice versa, both in healthy individuals [29, 75, 76] and in patients with mental disorders [44]. Notably, Hirot et al. [77] assessed self-recognition abilities in patients with AN via such a randomized self/other face morphing task using static images. They showed a significative difference with healthy subjects, with AN participants having more difficulty detecting facial changes and requiring images containing more "self" for self-recognition. The authors concluded to impairments in the recognition of one's own face, a marker of self-awareness, in patients with AN. Furthermore, contrary to the previously suggested hypothesis of a predominant egocentricity in AN, these results instead support an altercentric predominance in the SOD, with a greater tendency to recognize the other rather than oneself.

However, it can be questioned whether static image recognition paradigms are sufficient for SOD assessment. Indeed, several studies have supported that the sense of one's own body, variously termed "embodiment" [78] or bodily self-consciousness [79] is increasingly relevant for understanding clinical conditions such as AN [80, 81]. Therefore, studies of AN from an embodied perspective are urgently needed, as well as paradigms involving participants who are physically present to improve our understanding of AN physiopathology [82].

Here, for the first time, we examined self-recognition and SOD abilities in individuals with AN using a self-versus-other face identification task through a novel paradigm using the Double Mirror "Alter Ego"TM invented by the artist Moritz Werhmann.

This new paradigm developed by Berthoz and Thirioux allows us to specifically explore SOD under greater ecologically relevant conditions than the use of static images by merging the faces of two subjects physically facing each other and interacting on two sides of the mirror. It has been successfully used in several studies on self/other interactions in healthy subjects [83] and patients with psychiatric disorders, especially schizophrenia [45] and Autism Spectrum Disorders [84]. As shown in Fig 1, participants watch a double mirror in which a picture of their own face gradually transforms into the face of an unfamiliar other (self-to-other sequence) or vice versa (other-to-self direction) and indicate at which point they judge the morph to look more like their own face than the other person's face.

A mirror is a familiar and ecologically relevant everyday tool that plays a crucial role in several theoretical systems, from both developmental and psychodynamic perspectives. An ability for self-recognition in a mirror is indicative of an underlying self-concept and an important behavioural marker of higher-order consciousness [85, 86]. In particular, the ability to differentiate one's own image from that of another in the specular image is considered to be a precursor of self-awareness reached at approximately 4 months old [87]. Moreover, Zazzo [88]

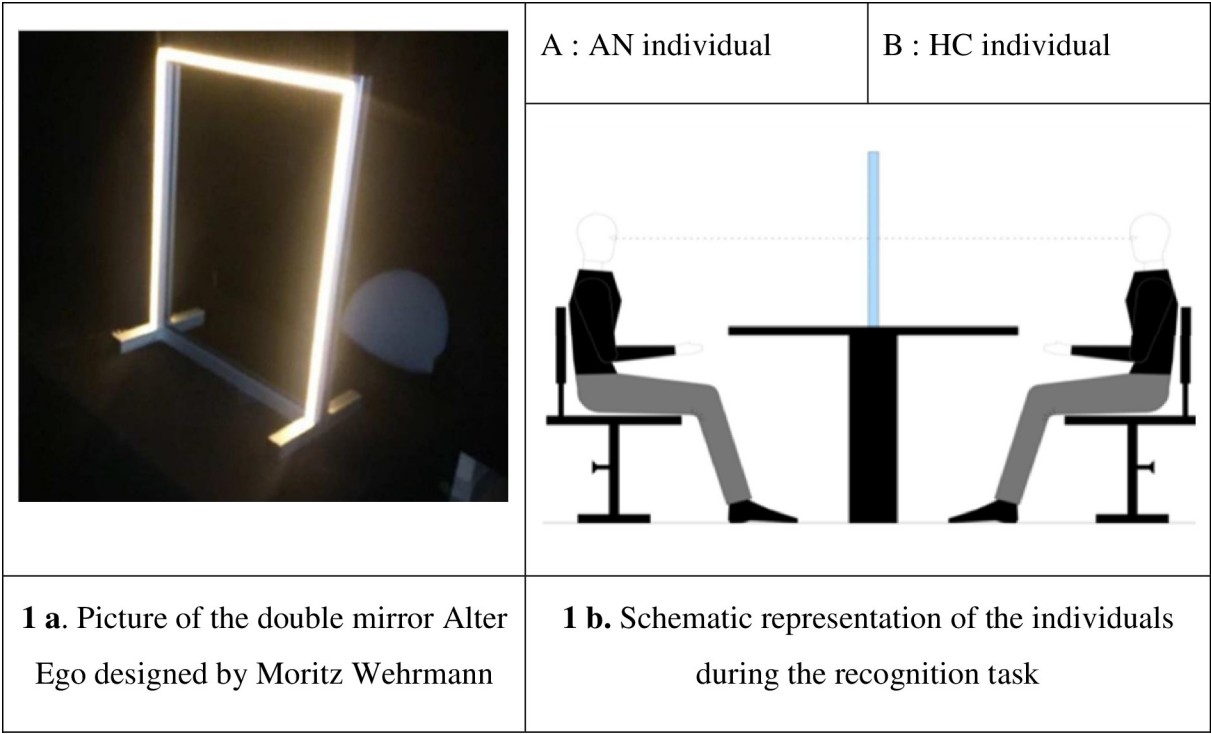

| | A : AN individual | B : HC individual |
|---|---|---|
| **1 a.** Picture of the double mirror Alter Ego designed by Moritz Wehrmann | **1 b.** Schematic representation of the individuals during the recognition task | |

**Fig 1.** Experimental setting (Fig 1a + Fig 1b). **Fig 1a. Picture of the double mirror Alter Ego designed by Moritz Wehrmann.** The experiment took place in a completely darkened enclosed space in the research centre of the Brest Hospital. The mirror was installed on a square table. The AN patient and his matched control sat facing each other on either side of the double mirror.
**Fig 1b. Schematic representation of the individuals during the recognition task.**
Both participants wore a black cape covering the neck, the whole chest and the arms so that only the face was visible. They were asked to look each other straight in the eye and to focus only on the other person's face. Height-adjustable chairs allowed strict alignment of the eyes.

described the way in which recognition of others (acquired at 8 months) far precedes self-recognition (acquired at approximately 2 years old) with the progressive awareness of one's own body image. From a phenomenological perspective, the image reflected in the mirror also reveals the ambiguity of realizing one's body as an object visible from the outside or by others, and yet being that object oneself—in other words, a tension between a primary embodied first person perspective and an internalized third-person perspective [89]. This intrapersonal tension also becomes an interpersonal one: while realizing that I am objectified or even reified by the gaze of the other, I must resist this gaze and (re)assert my own subjectivity, even if it is by objectifying the other in turn [90].

The objective of the present study was to examine self-recognition and SOD abilities in adolescents with AN compared to Healthy controls (HCs) by using the ability provided by this device to progressively change the identity of self and other in the mirror with changes in lighting on both sides of the device. This is a special kind of morphing using real actors, in contrast with the computerized methods usually used in such studies [29, 44, 75, 76].

Furthermore, in order to explore the relationships between body schema distorsions and impairments in social interaction and cognition, different sensorimotor conditions were applied to examine their effects on self-recognition and SOD.

Thus, by considering face recognition in a mirror as a key marker of self-consciousness [85, =86] but also the crucial first interface between self and other [91], our study aims to explore an interpersonal dimension of self-disturbance in AN, which is still lacking in the literature.

## 2 Material and methods

### 2.1. Participants

The study was conducted with seven individuals with AN (mean age: 15.6 years; 1 male, 6 females) matched by age and sex with seven unfamiliar Healthy control (HC) individuals. For logistical reasons related to the availability of the device, we were not able to recruit more participants at this stage of the research. However, given that small samples may be sufficient to show the presence of an effect but not for estimating the effect size [92, 93], this limitation is compatible with the hypotheses of this pilot study; specifically, that independent of the statistical representativeness of the study, AN individuals would exhibit differences in SOD compared to HCs on semiecological Double Mirror paradigm. Nevertheless, we successfully matched our participants with respect to age and age at testing.

Individuals with AN were recruited from Brest University Psychiatric Hospital in a unit specialized in the treatment of AN. None of the patients suffered from acute symptom exacerbations at the inclusion. They were all stabilized at recruitment and testing and not were taking any medication.

The inclusion criteria were (a) a diagnosis of AN made by a psychiatrist according to the DSM-V [94] and ICD-10 [95] criteria; (b) aged between 12 and 17 years old to ensure that a stable body schema had been acquired [96]. We excluded individuals with (a) a history of epilepsy, (b) claustrophobia, (c) achluophobia, or (d) the presence of distinctive signs or visual deficits requiring vision correction (eyeglasses or contact lenses).

The clinical characteristics of participants with anorexia are reported in Table 1.

The HC individuals matched with the AN patients were recruited in the Brest area by oral communication with local schools via participants and staff working at the hospital. They were unfamiliar with individuals in the AN group to avoid confusion between self-identification and familiarity. They were determined to (1) be free of any significant developmental, neurological or psychiatric disorders based on a medical examination nor personal history of eating disorders (2) have a normal BMI greater than or equal to 18.5 to 24.9 kg/m^2(3) attend typical schooling for which their chronological age corresponded to their developmental age. Similar to the AN group, the exclusion criteria for the HCs were (1) a history of epilepsy, (2) claustrophobia, (3) achluophobia, and (4) the presence of distinctive signs or visual deficits requiring vision correction (eyeglasses or contact lenses).

The study protocol was approved by an ethical standards committee and performed in accordance with the ethical standards laid down in the Declaration of Helsinki. Written informed consent was obtained from all participants and their parents prior to their inclusion in the study.

**Table 1. Clinical characteristics of participants with anorexia nervosa.**

| Participants | Gender | Age (years) | Anorexia Subtype | Body Mass Index (kg/m2) | Laterality |
|---|---|---|---|---|---|
| A1 | F | 15 | AN-R | 17.27 | RH |
| A2 | F | 15 | AN-R | 17.64 | RH |
| A3 | F | 13 | AN-R | 16.8 | RH |
| A4 | M | 16 | AN-R | 17.82 | LH |
| A5 | F | 17 | AN-R | 17.07 | RH |
| A6 | F | 16 | AN-R | 16.78 | RH |
| A7 | F | 16 | AN-BP | 24.22 | RH |

F: female, M: male, AN-R: anorexia nervosa restrictive subtype, AN-BP: anorexia nervosa binge purge subtype, RH: Right-handed, LH: Left-handed

## 2.2. Paradigm

In this experiment, we used a new double mirror paradigm based on the Alter Ego System, which consists of a semitransparent double mirror (70 cm × 50 cm × 0.4 cm; height×width×-depth) with a set of computer-controlled white light-emitting diodes (LEDs) fixed on the mirror frame on both sides (Fig 1A).

These sets of LEDs can emit continuous lighting at different intensities, either separately (i.e., LEDs turned on for only one side of the mirror) or simultaneously (i.e., LEDs turned on for both sides of the mirror). Both the sampling switch between the two LED sets and the flicker frequency range (1–20 Hz) were controlled by a PC using E-Prime software. This system enables the generation of different self-face and other-face perceptual conditions when two individuals, A and B, are facing either side of the mirror (Fig 1B).

- If the LEDs are turned on for subject A's side, whereas the LEDs are turned off for subject B's side, subject A can see his or her own face reflected in the mirror without seeing subject B's face through the mirror. This perceptual condition is referred to by Thirioux [83] and Keromnes [45] as the *self condition*.

- Using this same lighting mode, subject B can see subject A's face through the mirror (through a transparent window) without seeing his or her own reflection. This perceptual condition is referred to as the *other condition* [45, 83].

- When both sets of LEDs are on, the reflections of subject A's and subject B's faces merge in the mirror, making it potentially difficult for an individual to recognize his or her own face. The higher the light intensity is, the more visible the image of an individual in the mirror.

  The experimental setting is described in Fig 1.

## 2.3. Protocol and task

**2.3.1. Procedure.** The experimental procedure had a duration of approximately 90 minutes and was divided into 2 parts:

- First, the *neutral condition*, lasted 15 minutes and consisted of a « back-and-forth » passage, i.e. TDCs moved progressively from self to other, then from other to self while, during the same period of time, participants with AN experienced the opposite condition, i.e. they moved from other to self, then from self to other;

- The second part, called the *sensorimotor stimulation condition*, lasted 60 minutes and involved exposing participants to the same light intensity conditions as described in the neutral condition, i.e. a "back-and-forth » passage, but repeated 5 times with a different stimulus applied to the participants with AN during each passage.

  Thus, the whole task consisted of 6 back-and-forth passages: one in neutral condition, then 5 with different successive sensory stimuli only for AN participants.
  There was a 10-min pause between the first and second parts to allow attentional recovery.
  *First part*: *Neutral condition*. This experimental protocol was inspired by the one previously used by Du Boisgueheneuc in Alzheimer's disease patients, Keromnes and Tordjman in patients with schizophrenia [45] and Lavenne-Collot in patients with ASD [84]: the light intensity of the two LED sets were gradually and independently increased or decreased on the two sides of the mirror, such that the two individuals found themselves alternatively in the *other condition* or the *self condition*, as explained below.
  Before starting the task, participants were seated on opposite sides of the device so that the eyes of both partners were strictly aligned in the mirror reflection. They were instructed not to

make any body or facial movements (including facial expressions or grimaces) for the whole duration of the task.

At the beginning of the experiment, the patient was in the *other condition*, i.e., he starts by seeing the control subject's face through the mirror without seeing his own face; then, the patient's own image in the mirror becomes increasingly more apparent as a function of the light intensity.

Then, the patient is in the *self condition*, i.e., he begins to see his own face reflected in the mirror without seeing the control's face; then, with changes in the light intensity, the control subject's image in the mirror becomes increasingly more apparent.

At the same time, the control subject undergoes the same procedure, except that he starts in the *self condition* which transforms into the *other condition*.

Similar to Keromnes et al. [45], we called a "passage back and forth" the series of conditions that permitted each subject to return to the starting conditions after the conditions with identical light intensities was presented twice. Thus, one passage back and forth consists of the following sequence:

| One passage back and forth | { | TDC progressively switch from *self to other* and then *other to self* |
|---|---|---|
| | | AN simultaneously switch from *other to self* and then *self to other* |

The simultaneous alterations in light intensity on each side of the mirror experienced by the AN patient and his matched control are presented in Fig 2A. Importantly, since modifications in the light intensity on one side of the mirror modify its reflective properties on both sides, we should consider the relative light intensity that is actually perceived by participants on each side, as shown in Fig 2B.

After every change in light intensity, the participants were instructed to identify whether the presented image mostly corresponded to their own face or to the other's face. Both subjects were asked a simple question: "who do you recognize most in the mirror?" The expected response was either "me" or "her/him" (to designate the other person), without any other alternative (i.e., there were no different answers and no answer was not permitted). The question was always addressed first to the individual with AN and then to the HC participant. Each stimulus was presented for 10 seconds each during a progressive sequence. The response time was unlimited. Once the answer was given by each participant, the next stimulus was presented. The type of response measured and recorded was a verbal response. The use of response button boxes was initially discussed as a way to control for bias in participants' verbal responses. However, in addition to the difficulty of using manual buttons in totally dark lighting conditions, visual control might have been required to press the button, and thus, attention to the mirror image could have been disrupted or the movement could have induced a misalignment of the 2 faces. In addition, as in Keromnes [45], oral responses seemed to reassure these patients, as the task involved maintaining relationships with others and were more relevant in terms of identity reinforcement and assertive self-recognition.

*Second part*: *Sensorimotor stimulation condition*. In this second part, the same procedure was applied i.e. each stimulus of identical light intensity was presented twice to the participants during one passage back and forth as previously described in the *neutral condition*, but the postural or sensory stimulation conditions for the AN individuals were changed while the configuration for the HC individuals remained unchanged throughout the whole procedure. Indeed, our aim was to explore whether sensorimotor conditions reinforcing body image could improve the performance obtained by the AN participants. Accordingly, the tasks were

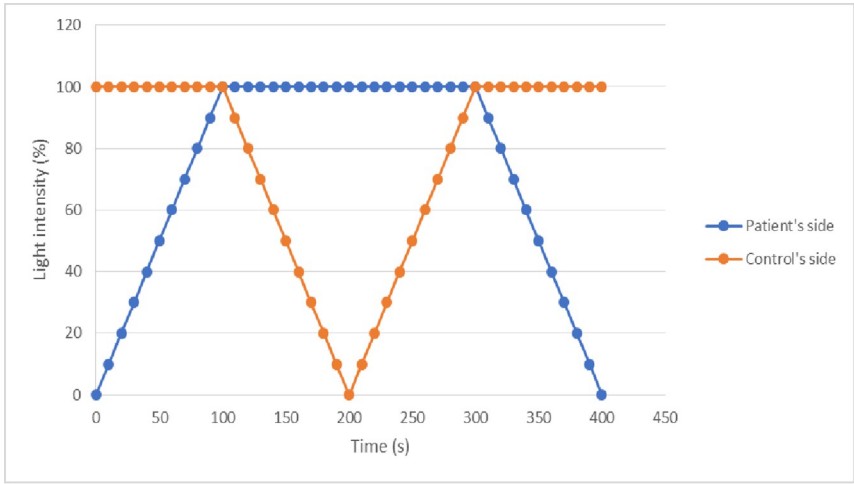

**a Light intensity over the time**

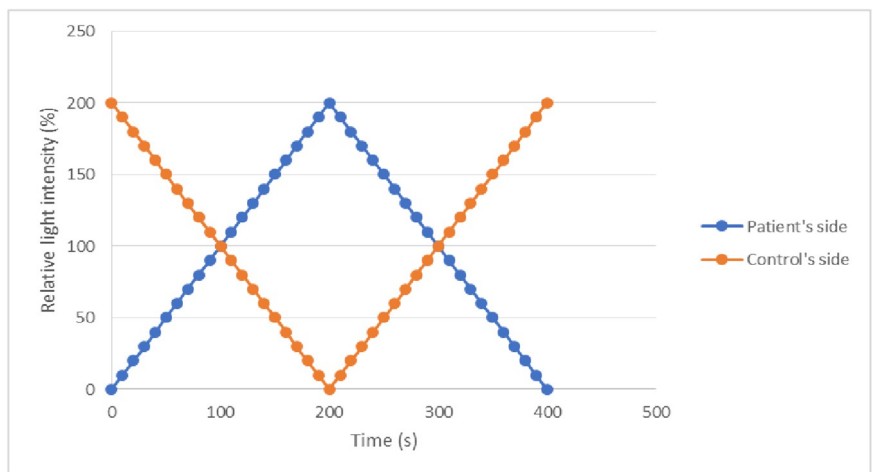

**b Relative light intensity perceived by each participant over time**

**Fig 2.** Simultaneous alterations in light intensity on each side of the mirror over time (Fig 2a + 2b).

**Fig 2a. Light intensity over time**.- At the beginning of the procedure, the light intensity was 100% for the HCs (mirror effect), whereas the light intensity was 0% for the AN patients (transparent window effect).

-Then, the light intensity is progressively increased in 10% steps on the AN patient's side while remaining at 100% for the HCs until 100% is reached on both sides.

- During this first step, the AN patient's image progressively appears and becomes increasingly fused with the HC's image.

- When both sides are at 100%, the light intensity is progressively decreased in 10% steps on the HC's side, while it remains at 100% on the AN patient's side.- During this step, the HC's image progressively fades until its total disappearance when the light intensity drops to 0%.

- Then, the reverse procedure is used to return to the initial configuration (100% on the HC's side, i.e., a mirror effect for the HCs, and 0% on the AN patient's side, i.e., transparent window effect for the AN patients).

- Therefore, a condition with identical light intensities is presented twice to the participants during this sequence called one "passage back and forth".

**Fig 2b. Relative light intensity perceived by each participant over time.**

Relative light intensity perceived by the participants taking into account that modifications of light intensity on one side modifies its reflective properties on both sides.

used only for AN participants to investigate this compensatory effect, assuming that healthy control participants have intact bodily self-consciousness abilities (without the need for reinforcing supports)

As previously mentioned, body schema and self consciousness are both based on a multisensory integration process [8]. Therefore, we modified sensory inputs in participants with AN to assess their effect on mirror self-recognition and SOD. Five different tasks were administered, including different sensory modalities and locations, in agreement with psychomotor therapists, based on their knowledge of strategies used to reinforce body schema in usual care of patients with AN.

The objectives of these different tasks were: (1) to increase the awareness of each of the 2 hemi-bodies by successive weighting of its (1) right and (2) left sides, (3) to reinforce the median axis of the body by providing increased postural support at the level of both the back and the pelvis, (4) to stimulate the vestibular system with a cushion generating an unstable position, and (5) to provide intensive somatosensory stimulation with a neoprene jacket.

Participants were tested successively on each of these distinct tasks that were administered always in the same order:

1. **Right hemi-body weighting (RHB) condition**: Weights of 75 grams were placed around the AN patient's right wrist and right ankle

2. **Left hemi-body weighting (LHB) condition:** Weights of 75 grams were placed around the patient's left wrist and left ankle

3. **Back- and Foot- Rest (BFR) condition**: Patients with AN were no longer seated on a stool but in a chair with a backrest and a step was placed under their feet to allow pelvic flexion

4. **Unstable Cushion (UC) condition:** Patients with AN were sitting on an unstable cushion installed on the stool

5. **Neoprene (NP) condition**: Patients wore a neoprene jacket

Debriefing occurred at the end of the experiment during an informal and friendly period of time that did not include data collection. A snack with cakes and drinks was offered during this period. The participants were asked about their feelings regarding the task and their awareness about differences in stimulation, asked whether they found face recognition and discrimination difficult, and whether they were familiar with this type of morphing (as some teenagers regularly use morphing applications).

**2.3.2. Data analysis.**   The main outcome was the light intensity levels at which critical changes in self/other identification occurred:

1. Level M1 was the critical perceptual threshold corresponding to the first time the individual recognized himself when his own image progressively appeared in the mirror during the *other-to-self* morphing sequence.

2. Conversely, level M2 was the critical perceptual threshold corresponding to the first time the individual recognized the other's image in the mirror when this image progressively appeared in the mirror during the *self-to-other* morphing sequence.

To summarize, the lower the M1 or M2 levels were, the larger the proportion of self was in the image (and the smaller the proportion of the other), which reflected a difficulty for the individual to uncouple from his own image.

Determination of M1 and M2 levels as well as a synthesis of the experimental procedure are shown in Fig 3.

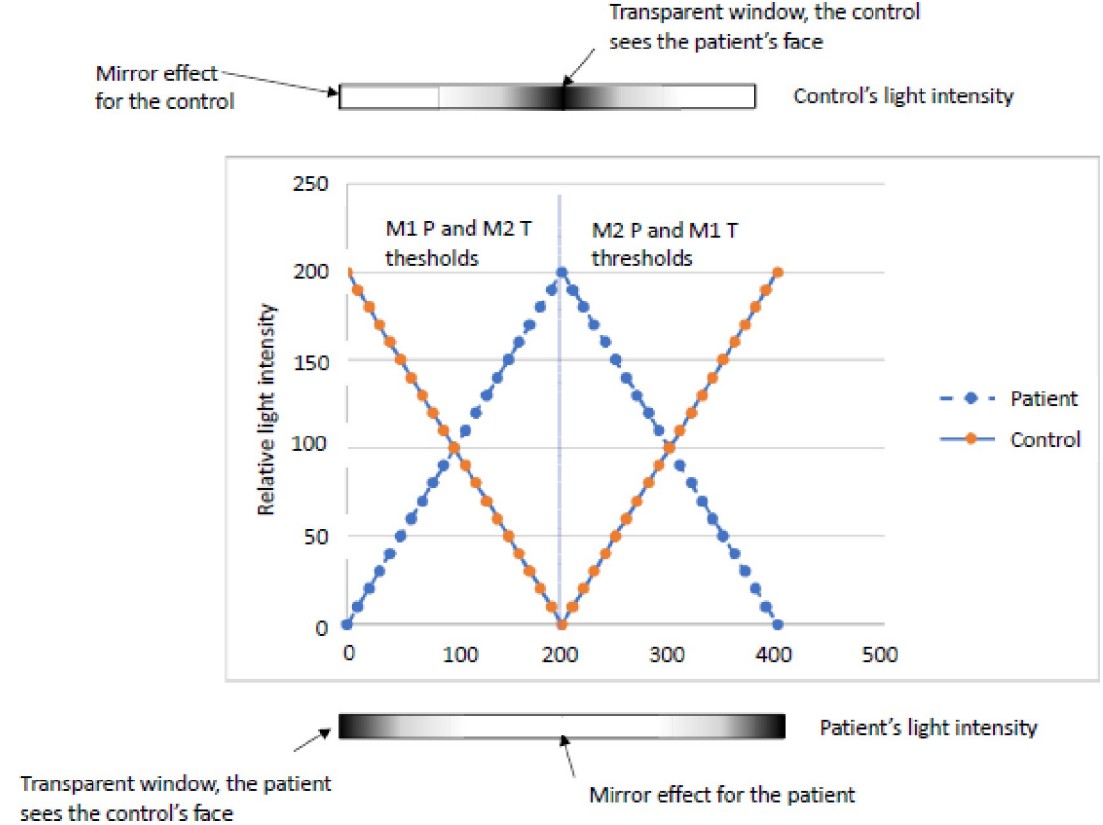

**Fig 3. Synthesis of the experimental procedure during one "passage back and forth".** During the task, the image the AN patient sees undergoes a morphing from other to self and then from self to the other. Simultaneously, the image the HC sees undergoes a morphing from self to other and then from other to self. The threshold M1 corresponds to the light intensity at which the perceptual shift from other to self occurs. The M2 threshold corresponds to the light intensity at which the perceptual shift from self to other occurs. The procedure is then repeated a second time. Thus, for each subject, 2 M1 and 2 M2 thresholds are obtained.

**2.3.3. Statistical analysis.**   After each step (change in light intensity), the participant's verbal responses were recorded. The analysis of the main outcome variable was conducted by comparing the M1/M2 levels (expressed as percentage of light intensity) between individuals with AN and HCs. As this variable was not normally distributed, the nonparametric one-tailed Mann–Whitney test was used to compare M1 and M2 levels between the two groups.

The effects of differents sensorimotor tasks on the M1 and M2 levels were assessed by using the non-parametric Wilcoxon test to compare for the AN patients the results obtained in the neutral condition and the results obtained in the 5 sensorimotor conditions (RHB, LHB, BFR, UC, NP).

## 3 Results

The comparison of the results for the self/other distinction task in the different conditions between individuals with AN and HCs are presented in Table 2.

### 3.1. Neutral condition

In neutral condition, M1 levels were significantly lower in the individuals with AN than in the HCs (p = 0.001). This indicated that the AN individuals showed an "earlier" self -recognition

**Table 2. Comparison of the results for the recognition task between individuals with AN (N = 7) and Healthy controls (N = 7) in Neutral (Neu), Right hemi-body weighting (RHB), Left hemi-body weighting (LHB), Back- and Foot- Rest (BFR), Unstable Cushion (UC) and Neoprene (NP) conditions.**

| | Individuals with AN | | | Healthy controls (HCs) | | | |
| | (N = 7) | | | (N = 7) | | | |
| | Mean | Median value | SD | Mean | Median value | SD | P-value |
|---|---|---|---|---|---|---|---|
| | | | | | | | |
| M1 Neu | 77.1 | 70 | 18 | 132.9 | 130 | 17.0 | **0.001** |
| M1 RHB | 84.3 | 80 | 22.3 | 137.1 | 140 | 29.8 | **0.004** |
| M1 LHB | 87.1 | 80 | 26.9 | 128.6 | 140 | 21.9 | **0.004** |
| M1 BFR | 107.1 | 90 | 32 | 125.7 | 140 | 25.7 | 0.149 |
| M1 UC | 80 | 80 | 20.8 | 135.7 | 140 | 27 | **0.004** |
| M1 NP | 84.3 | 90 | 15.1 | 130 | 140 | 22.3 | **0.003** |
| M2 Neu | 135.7 | 130 | 11.3 | 150 | 150 | 10 | **0.021** |
| M2 RHB | 117.1 | 110 | 18.9 | 141.4 | 150 | 12.1 | **0.012** |
| M2 RHL | 118.6 | 120 | 15.7 | 141.4 | 140 | 9 | **0.004** |
| M2 BFR | 124.3 | 130 | 18.1 | 128.6 | 130 | 17.7 | 0.277 |
| M2 UC | 111.4 | 110 | 10.7 | 140 | 140 | 11.5 | **0.001** |
| M2 NP | 111.4 | 110 | 10.7 | 135.7 | 130 | 11.3389 | **0.003** |

in the *other condition* (i.e., when one's own image was gradually appearing in the mirror in the other-to-self direction of the morphing).

Similarly, M2 levels were significantly lower in the AN individuals than in the HCs (p = 0.02). This indicated that the AN individuals showed a delay in recognizing the other in the *self condition* (i.e., when the image of the other was gradually appearing in the mirror in the self-to-other direction of the morphing).

The Wilcoxon paired-samples test comparing M1 and M2 showed that M2 levels were significantly higher than M1 levels in individuals with AN (p = 0.02). Conversely, there was no significant difference between M1 and M2 levels in HCs (p = 0.09). This indicated that, in the HCs, the critical perceptual threshold corresponding to the ability to recognize self and other was the same regardless of the direction of morphing. In other words, the critical threshold for switching between self and other appeared when viewing morphed faces that contained the same proportion of facial features of self and other, regardless of whether it was the *other* condition or *self* condition. In contrast, in the individuals with AN, this perceptual threshold varied depending on the direction of the morphing, and the shift from self to other or from other to self did not occur at the same threshold. Indeed, the AN group needed a smaller proportion of self (image of themselves) to recognize their own face in the other-to-self direction versus the self-to-other direction.

Comparison of M1 and M2 levels in ANs and HCs are presented in Fig 4 and a synthesis of M1 and M2 distribution levels in neutral condition is presented in Fig 5.

## 3.2. Sensorimotor stimulation condition

When the AN individuals were installed in the LHB, RHB, UC and NP conditions, M1 levels were significantly lower in the AN individuals than in the HCs. This indicated that, in all those conditions, AN individuals showed an earlier self- recognition in the *other condition* (i.e., when one's own image was gradually appearing in the mirror in the other-to-self direction of the morphing).

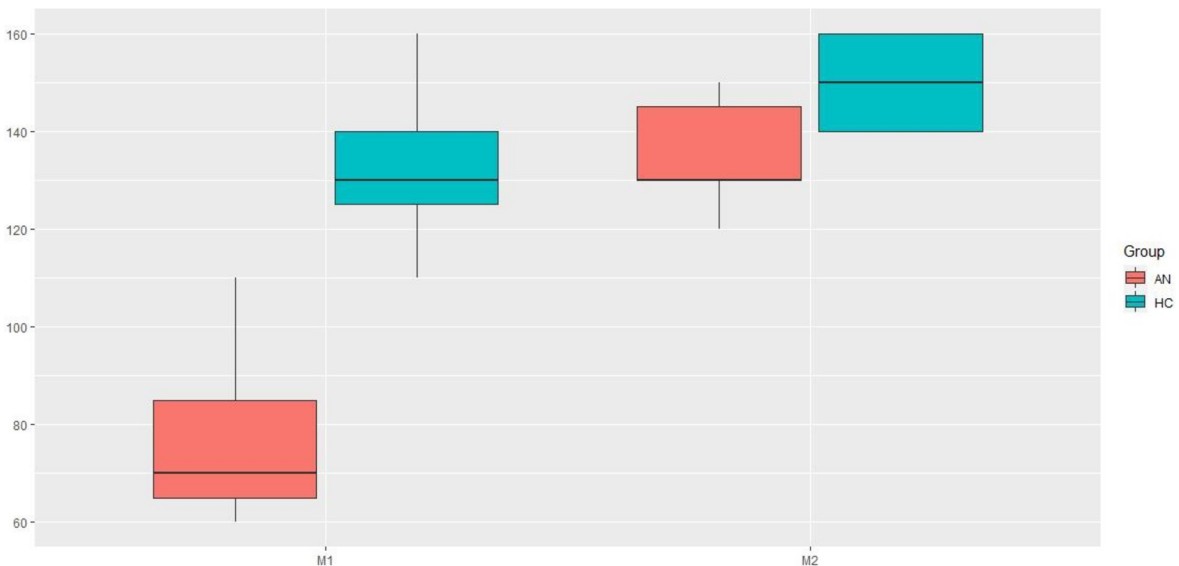

**Fig 4. Comparison of M1 and M2 levels in HCs and ANs in neutral condition.** This figure indicates that, in the HCs, the critical perceptual threshold corresponding to the ability to recognize self and other was the same regardless of the direction of morphing. In contrast, in the individuals with AN, this perceptual threshold varied depending on the direction of the morphing, and the shift from self to other or from other to self did not occur at the same threshold.

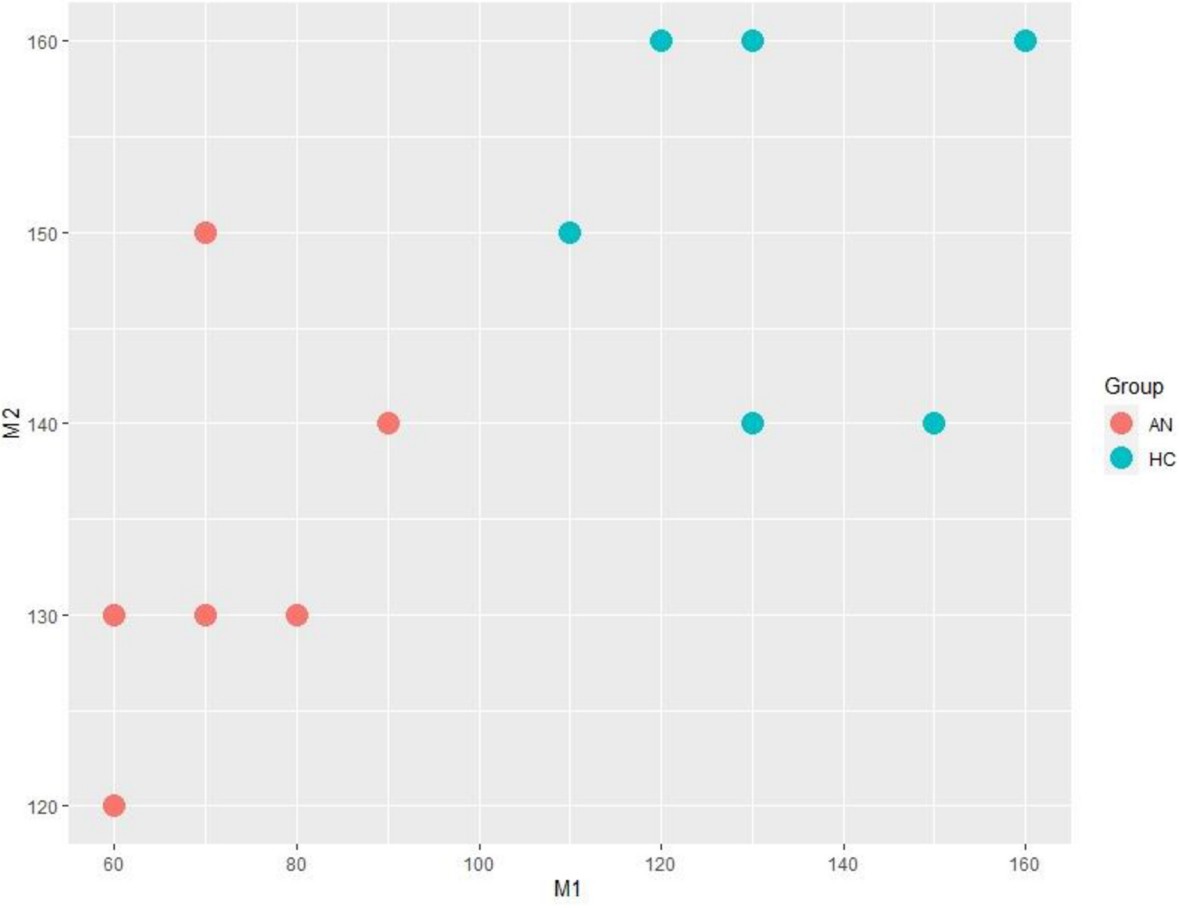

**Fig 5. Distribution of M1 and M2 in AN and HC in neutral condition.** Two points are superimposed.

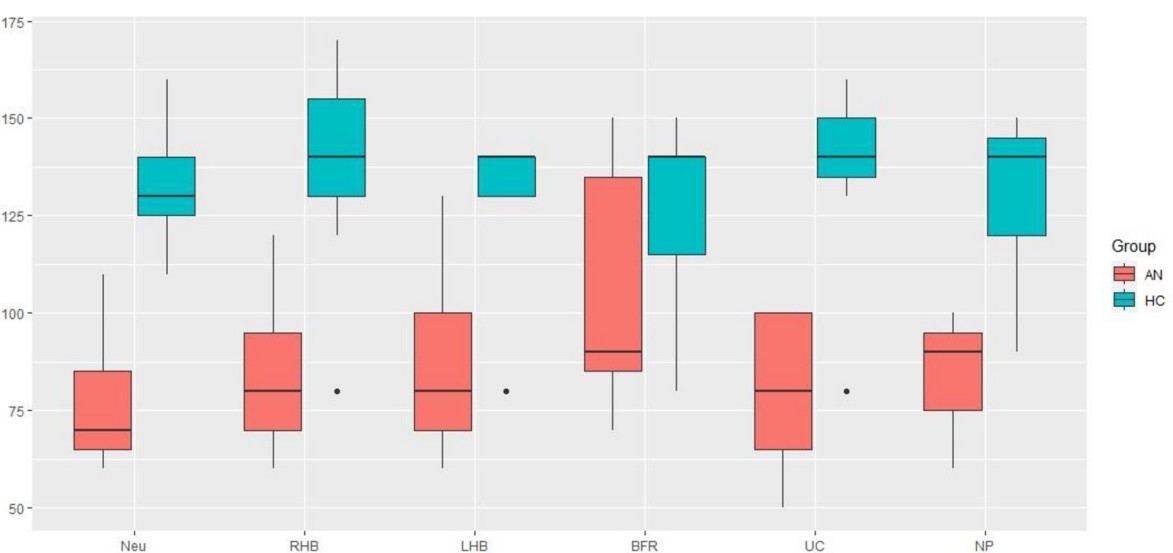

**Fig 6.** Distribution plots of M1 levels in AN and HC in Neutral (Neu), Right hemi-body weighting (RHB), Left hemi-body weighting (LHB), Back- and Foot- Rest (BFR), Unstable Cushion (UC) and Neoprene (NP) conditions.

Conversely, there was no significative difference on M1 levels in ANs compared to HCs in the BFR condition. This indicated that, when AN individuals were seated in a chair with a backrest and a support under their feet, there was no significant difference in self-recognition levels compared to HCs. Distribution plots of M1 thresholds for ANs and HCs in the different conditions are presented in Fig 6.

When the AN individuals were installed in the LHB, RHB, UC and NP conditions, M2 levels were significantly lower in the AN individuals than in the HCs. This indicated that, in all those conditions, AN individuals showed a delay in recognizing the other in the self condition (i.e., when the image of the other was gradually appearing in the mirror in the self-to-other direction of the morphing).

Conversely, there was no significant difference in M2 levels between ANs and HCs in the BFR condition. This indicated that, when AN individuals were in the BFR condition, there was no significant difference in other- recognition level compared to HCs.

Distribution plots of M2 levels in different conditions for individuals with AN and HCs are presented in Fig 7.

To summarize, in the AN patients, M1 and M2 levels were significantly lower than in HCs in neutral, LHB, RHB, UC, and NP conditions, indicating that patients with AN recognized their own image earlier when it appeared in the mirror in the *other* condition and that they recognized the other person's image later when the other was gradually appearing in the mirror in the self condition.

Conversely, there was no difference between ANs and HCs on the M1 and M2 levels when the AN individuals were in the BFR condition.

### 3.3. Comparison between neutral and sensorimotor conditions

In the AN group, M1 levels were significantly higher in the BFR than in the neutral condition. Conversely, there were no significative differences on M1 level between neutral and LHB, RHB, UC and NP conditions.

In the AN group, M2 levels were significantly lower in the LHB, RHB, UC, and NP conditions than in the neutral condition. Conversely, there was no significant difference on M2 levels in the BFR condition compared with the neutral condition

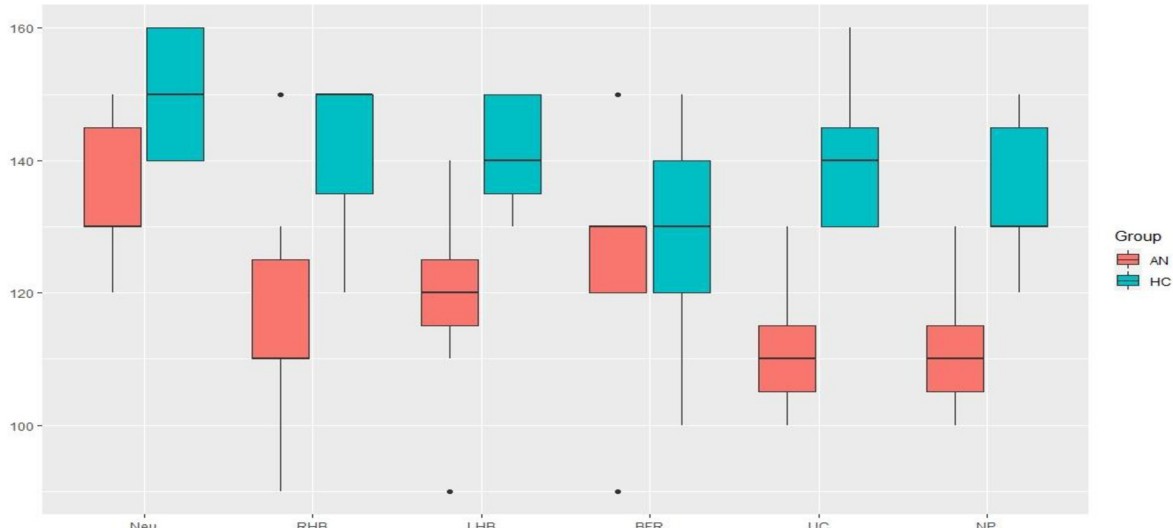

**Fig 7.** Distribution plots of M2 levels in AN and HC in Neutral (Neu), Right hemi-body weighting (RHB), Left hemi-body weighting (LHB), Back- and Foot- Rest (BFR), Unstable Cushion (UC) and Néoprene (NP) conditions.

These results indicated that when the recognition task was associated with the BFR condition versus the neutral condition, AN participants showed a delay in self recognition but no significant difference in other recognition. Conversely, in ANs, there was a delay in recognition of the other when the visual recognition was associated with the LHB, RHB, UC, NP conditions compared to the neutral condition, but no significant difference in self recognition.

## 4 Discussion

Our results are based on a very small sample; therefore, the preliminary nature of this study must be taken into account when analysing and discussing the results.

### 4.1. Self recognition in the mirror

**4.1.1. Self- consciousness.** The results of the present study support that AN individuals show a difference in mirror face recognition thresholds compared to HCs on the double mirror task. These results are consistent with previous research showing that self-recognition abilities may be impaired in AN [59, 77]. Furthermore, in line with previous research showing the value of the double mirror device in studying bodily self-consciousness [46, 97], our results support the notion that self-disorder is an important dimension of AN [98, 99] and, in particular, that the embodied aspects of the self could be disturbed [5, 100].

**4.1.2. Comparison with previous research using static images.** Our results showed that the AN individuals required images that contained less proportion of self than controls to recognize themselves in the reflected image, regardless of the direction of the morphing (i.e., in the "*self*" or "*other*" conditions). These results contrast with those found by Hirot et al. [77] during a self/other morphing task using static images that showed that patients required images containing more "self" to recognize themselves. Several arguments support this discrepancy: First, (i) contrary to photo, looking at ourselves in the mirror gives us access to our own image, besides the proprioceptive, tactile and motor sensory cues that are necessary for the representation of one's own face [101]; (ii) studies have found that mirror self-recognition emerges prior to photo self-recognition [102]; (iii) different neural responses have been found when comparing mirror and photo self-processing [103] and; (iv) preserved self-face

recognition on photographs despite incapacity in mirrors has been described in some neuro-logical patients showing a kind of agnosia for recognition of their own reflected image [104]. Moreover, our experimental design differs from that of Hirot's study because they used an aleatory morphing protocol thought we presented the morphs incrementally from 0% self/other to 100% self/other in two directions separately.

## 4.2. Egocentricity in self/other distinction

**4.2.1. Implications regarding spatial cognition: Privileged egocentric reference frame.** The results obtained in the present study showing earlier self-recognition and delayed other recognition in the AN individuals, compared to the HCs, could be interpreted as a difficulty to inhibit their own perspective during the face recognition task and adopt the reference frame of the other, resulting in a privileged use of the egocentric reference frame.

Indeed, unlike when one is faced with a simple photograph, the mirror requires processes of perspective shifting and spatial transformations beyond pure self-recognition. In particular, self-recognition in the mirror requires matching one's sensorimotor experience (1st person perspective) with the object seen in the mirror (3rd person perspective), thereby identifying the "I" with the "me" and representing that self as an object to others and to oneself [103]. Moreover, the face-to-face postural configuration of both subjects during the Alter Ego double mirror task requires a 180˚ mental rotation of one's own body. In particular, Thirioux et al. [83] emphasized the role of visuospatial abilities on self-recognition and SOD within this task. Thus, our results are consistent with previous studies showing alterations in spatial cognition in ANs and, in particular, a privileged use of the egocentric frame of reference during spatial performance tasks [14, 55]. Moreover, given the implications of impaired spatial references for social cognition and perspective shifting or empathy [54, 56–58], this raises the question of the ability of patients with AN to infer mental states other than their own. To date, the nature of the links between alterations in body schema and alterations in spatial cognition remains unknown [14]: is it the alteration of the body signal that may participate in the disruption of spatial cognition (bottom up effect)? Or is it the alteration of spatial cognition that may participate in the alteration of body schema (top down effect)? Overall, analysis of the results under sensorimotor conditions tends to support the second hypothesis (see below).

**4.2.2. Implications for multisensory processing: Overinclusion of the other's face.** Our results showed that the AN individuals, compared to the HCs, judged the morphs to be more like the self in both directions of morphing. These findings can be interpreted as a greater level of assimilation of the other's face in the representation of one's own face among the AN individuals than the HCs. Such misattribution of others' facial features to oneself has been reproduced in nonclinical populations through simple psychophysical manipulations in a procedure involving sensory processing called interpersonal multisensory stimulation [29]. Indeed, studies using body ownership illusions have shown that, under certain conditions, the sense of self can be manipulated to include a fake or another person's body part, for instance, the hand [105], face [101, 106, 107] or whole body [108].

The double mirror task involves a simultaneous visual perceptual conflict between the visual input ("I see the other") and the proprioceptive experience ("I expect to see myself in the mirror"). This conflict is reinforced by the everyday experience we may have developed that, when we look in the mirror, we see ourselves and not another person. As specified by Thirioux [83], here we studied the effect of a unisensory (visual) conflict, whereas previous studies have typically studied the effect caused by multisensory (especially visual-tactile) illusions [29].

Several studies have revealed abnormalities in the multisensory integration process in patients with anorexia nervosa [9]. In particular, people with anorexia nervosa have been found to exhibit greater sensitivity to the Rubber Hand Illusion [105], indicative of a greater

susceptibility to appropriating and embodying an object external to their own body, i.e. a more plastic bodily self [13, 109]. Yet conflicting results have shown similar levels of sensitivity in full-body illusion tasks compared to controls [110, 111], but only when a highly emotionally valenced body part, such as the abdomen, was stimulated to promote the illusion [9].To our knowledge, this greater susceptibility to the embodiment illusion for body parts (contrasting with full-body) has never been explored at face level in individuals with AN, as assessed through paradigms such as the "enfacement illusion" [106, 112]. However, both body plasticity and face representation have been shown to rely on the same multisensory integration processes [8, 101]. Thus, the results of the present study are consistent with the generalization to the face of a greater tendency to the embodiment illusion for body parts in AN individuals.

Moreover, our results indicated that people with AN continued to recognize themselves in the mirror despite the increase in the proportion of the other person's face in the image, regardless of the direction of the morphing. These results may indicate that, in the AN individuals, the proprioceptive system would be less vulnerable to bias originating from visual information (i.e., AN individuals depend more heavily on proprioceptive information than on visual information when incongruent). This finding is in line with previous studies that pointed out an overreliance on proprioception in the presence of competing signals from other modalities [17]. Moreover, they are also consistent with reduced integration of visual and proprioceptive information in AN, also supported by the ubiquitous clinical finding in these patients that their visual perception of the body, especially in a mirror, does not correct their distorted body image [113].

**4.2.3. Comparison between the *self* and *other* conditions: Imbalance mechanisms of projection / simulation.** Our results showed that the AN individuals, compared to the HCs, judged the morphs to be more like the self in both directions of morphing. Importantly, the comparison between *self* and *other* conditions shows that, unlike the HCs, the AN individuals switched from *other to self* and from *self to other* at different thresholds. This discrepancy indicated that the confusion between self and other may have been determined by a unidirectional overinclusion of the other's attributes by the AN individuals. Conversely, the HCs may have exhibited compensatory mechanisms with bidirectional self- and other-facial feature attribution. i.e., they projected their facial features onto the other's face as much as they introjected their partner's facial features onto their own face. This understanding is supported by Thirioux's findings using a double mirror in healthy subjects [83] and human interpersonal understanding based on balanced mechanisms of self-projection and simulation [41, 114]. Indeed, as noted by Freud "the boundaries of the self are not constant" [115]. That is, both processes also transformed the margins between the self and the other, either by externalizing facial features of the self into the other (projection) or by alienating facial features that do not belong to the self but to the other (introjection) [116]. Thus, our findings may be consistent with psychoanalysts who described anorexia as a failure of identification and a pathology of object relations [117] particularly marked by deficiencies in the balanced introjection/projection mechanisms that are necessary to establish a sense of identity, leading to characteristic forms of pathological projective identification often evident in these patients [118].

**4.2.4. Switching between the abstracts of self and other.** Our experimental design requires switching from *self to other* and from *other to self* as the morphing gradually shifts from 0% to 100% between self and other and vice versa. Thus, the double mirror Alter Ego task is consistent with recent conceptualizations of SOD as being achieved through the ability to switch between representations of self and other and to inhibit the representation that is not relevant in a given situation [119]. In our study, differences in M1 and M2 thresholds between the AN individuals and HCs were consistent with AN being associated with an inadequate control of the self/other control or switch [37].

Another hypothesis explaining our results is that anorexic participants may have difficulties matching their own actions with their sensory inputs, i.e. difficulties in predicting the sensory outcome of their own action. This hypothesis is consistent with the theory of active inference, which postulates that the brain uses an internal model to continuously infer and control external or internal signals associated with the body [120, 121]. This theoretical framework is attracting growing interest in understanding alterations in sense of self [65], particularly in AN [122].

Indeed, although our task does not involve active movements, since participants were instructed not to move, we cannot exclude minimal movements such as eye blinking or respiratory movements. In any case, the double mirror task involves motor imagery, especially mental rotation [83] which implies predicting the sensory consequences of the imagined movement [123]. Usually, the brain weights and selects the most informative signals it receives to minimize prediction error and reduce surprise, notably by attenuating self-generated percepts that are highly predictable [124] through a copy of the motor signal called efference copying [125]. In AN, subjects have been shown to rely on incorrect predictions, i.e. on the unverified representation of their own body. In particular, some authors have highlighted a lack of integration of interoceptive and exteroceptive signals leading to an allocentric lock-in state, where body memory is not updated according to new sensory evidence relating to the body's actual shape [122, 126]. Interestingly, with regard to the distinction between self and others, prediction errors have been demonstrated at the tactile level in subjects with AN, the lack of neural attenuation of tactile sensations produced by self making them more similar to those produced by others [74]. This finding is particularly interesting considering the crucial role of tactile cues in the construction of self-consciousness from fetal life [63, 66, 67].

## 4.3. Impact of sensorimotor tasks on self-recognition and SOD

**4.3.1. Back and foot support condition.**    *Comparison between neutral and BFR conditions.* Our results showed a significant increase in the M1 threshold in the BFR condition compared to the neutral condition. This indicated that, when they were in the BFR condition, the AN participants showed a delayed self-recognition, i.e., they continued to perceive the other longer while their own image was gradually appearing.

Back support is an essential early developmental modality for the construction of one's identity, and in particular one's body. Indeed, during fetal development, the uterine wall provides the primary back support that allows for the development of oral activities (swallowing, etc.). Even after birth, back support remains fundamental, especially in the sensorimotor model developed by Bullinger [127] that argues that postural control and tonic-sensory balance allow infants to interact with their environment and develop social involvement [128]. Notably, in a pilot study, Rahme et al. showed a significant increase in social attention, in the back support condition compared to no back support in children with neurodevelopmental disorders [129]. Similarly, the results of our study are consistent with the idea that the BFR condition may have helped the AN individuals to be more other-centered.

*Comparison between AN and HC groups.* In contrast to the results obtained in the neutral condition, when the AN participants were settled in the BFR condition, there was no more significative difference between the HC and AN groups. These results suggest -that the implementation of postural supports against the medial axis would have compensated for the alterations in self-recognition and SOD in the AN participants.

**4.3.2. Other sensorimotor conditions.**    As far as other sensorimotor conditions are concerned, both left and right hemibodies ballasting, unstable cushion and neorprene conditions had a greater effect on M2 threshold than on M1, including a decrease in M2 threshold. This

indicates that these conditions had no significant effect on self-recognition, but were significantly associated with delayed other-recognition, suggesting that they may have helped the AN individuals to be more self-centered. These results are in line with previous research using double mirror in schizophrenia patients that showed that tactile and kinesthetic tasks helped to be more self-centered [45].

Furthermore, our results did not show any significant difference regarding the right- or left-lateralized tasks, which did not support a preferential input lateralization effect on face recognition and SOD thresholds in the AN individuals.

Regarding the Neoprene condition, previous research conducted in a single case study conducted by Grunwald showed that intensive somatosensory stimulation with a neoprene diving suit led to improved body image in the AN participants [130]. However, the neoprene suit was worn for one hour three times a day whereas, in our study, this condition may have been applied for too short a time to induce a change in self-recognition patterns.

## 4.4. Implications for AN comprehension: what's new?

### 4.4.1. Relationship between bodily self-consciousness and socio cognitive functioning.
In summary, our results highlight: (i) disturbances in mirror self-recognition suggesting bodily self-consciousness impairments in AN; (ii) an egocentric bias in self/other disctinction found in AN participants compared to HCs and; (iii) changes in self-recognition and SOD thresholds were observed when sensorimotor or postural conditions were modified in individuals with AN.

Although our preliminary results must be put into perspective given the small sample size, they support the hypothesis that more fragile bodily self-consciousness in AN participants may impair self-referential processes (bottom up effect). This hypothesis is consistent with the results obtained by Thirioux [83] who showed an alteration of self-referential processes by inducing self-fragility through self/other fusion via the double mirror in healthy subjects.

### 4.4.2. More fragile median body axis.
Our results support fragilities of the body schema at the level of the median axis in AN. This hypothesis is consistent with clinical observations showing, in these patients, a misalignment of the whole body, especially with respect to this axis, a restricted breathing pattern coupled with muscle rigidity having a major negative impact on body stability [131], problems with stabilization of the back and pelvis in a neutral position and difficulties in performing pelvic movements [132] as well as gait abnormalities with biomechanical differences primarily in the pelvis and hip [133]. Interestingly, while studies on the effects of body therapies aimed at increasing body-awarness (often referred to as 'body awareness therapies') are still rare, the authors of a recent study on the lived body in patients with anorexia found that by allowing themselves to lean on the seat and back of the chair (during physiotherapy consultations), instead of perching on the edge of the chair, anorexic subjects felt more attentive and present in the here and now, indicating an increase in self-consciousness through the addition of medial postural support, consistent with our findings [134].

Furthermore, our results did not find a preferential effect of lateralized tasks contrasting with previous research supporting similarities found in AN patients with hemineglect patients [135] but with divergent results, just as the existence of a hemispheric predominance is still debated [82, 135]. More than a hemineglect similar to the one found in neurological patients, our results may raise the hypothesis of a functional cleavage or split at the level of the median axis that is supposed to connect the right and left parts of the body.

### 4.4.3. Further therapeutic and research implications.
As they highlight the prominence of abnormalities in body self-consciousness in understanding alterations in social-cognitive and interpersonal functioning in AN, our results also support the value of physical treatments

in this condition. In particular, our results support that psychomotor therapy focused on strengthening and stabilizing the body axis could improve self-awareness, which could then have training effects on social-cognitive functioning.

In addition, the fragilities found in the median axis of the body raise the hypothesis of disturbances in the very early stages of body construction in AN patients, as observed in other neurodevelopmental disorders [127, 129]. Therefore, it would be interesting to look for possible early signs of vulnerability in the sensorimotor domain, following the example of retrospective studies using home movies performed for such disorders [136].

Finally, using the same protocol respectively in patients with schizophrenia and also ASD versus healthy controls, Keromnes and Tordjman [45] and Lavenne-Collot [84] found similar results to the present study (i.e., earlier recognition of self and later recognition of others). Interestingly, given the overlap in symptomatology between ASD and AN, it would have been interesting to collect participants' autism spectrum quotient (AQ) scores, but this goes beyond the scope of this preliminary study. Overall, these results underline the value of considering SOD disturbances as a transdiagnostic dimension shared with other psychiatric or neurodevelopmental disorders and supports the interest of further research using the Alter Ego in a double mirror at the diagnostic but also therapeutic level.

## Limitations and future directions

Some limitations of the study should be recognized. First, this is a preliminary study conducted with very small samples. Future studies are therefore needed to replicate the results in larger, samples.

Moreover, the self–other facial morphing task used in this study taps into perceptual SOD, that is, the capacity to identify one's own body (here, one's face) and to distinguish it from others. Therefore, it raises the question of whether the findings from the present study could be generalized to other domains of SOD. Although perceptual SOD and mental-state SOD should not be equated, evidence shows that they may be related [119] and that SOD may operate in a domain-general rather than domain-specific manner [137].

Another limitation of this study is that the differences between AN individuals and HCs may have occurred at the attentional or information processing level, i.e., upstream of the SOD process. Notably, our results could have been influenced by the cognitive functioning typically described in AN, including a deficit in cognitive flexibility, although these alterations showed contrary results in the literature (for a review see [138]). However, if a deficit in cognitive flexibility could account for the delay observed in the ability to switch from self to other that was found in the *self condition*, a similar delay should have been observed in the *other condition*, which is contrary to our results. Similarly, we don't think our results can not be explained by an attentional or emotional bias related to the fact that subjects with AN dislike seeing their own face in the mirror (i.e. they would have avoided or be more sensitive to seeing it). Indeed, if this hypothesis were correct, this tendency should have been the same regardless of the direction of morphing.this tendency should have been the same regardless of the direction of morphing.

In addition, patients with AN typically exhibit a strong desire for control that results in extreme organization and planning [139]. Given their usual perfectionism and fear of making mistakes, one might have expected a later self-recognition (similar to Hirot's [77] results), i.e., the presented images should have contained more of their own face for participants to be more certain before responding. Interestingly, again, our results are opposite.

Furthermore, in the second part of the study, the postural and sensorimotor tasks were only applied to participants with AN. Therefore, the effects of these tasks in healthy subjects remained unknown. However, here, the purpose of our study was (i) to explore the

relationship between body schema abnormalities and SOD abilities only in AN individuals (ii) to explore whether sensorimotor conditions reinforcing body image could improve the performance obtained by the AN participants. Therefore, the tasks were used only for patients to examine this compensatory effect compared to healthy controls.

Furthermore, we do not believe that the results obtained for the sensorimotor tasks can be explained by a distraction effect caused by the stimulation. Indeed, in this case, the distraction effect should have been observed in a similar way whatever the direction of morphing for the same stimulus. Moreover, the distraction effect should have been greater for the more unusual sensorimotor tasks (such as sitting on an unstable cushion). However, it is the effect of the BFR condition (which is not the most unusual) that emerges significantly from our results.

As far as experimental setting is concerned, there might be a bias of participants' oral responses influencing each other. To decrease this possible bias in the individuals with AN, these participants were consistently asked to respond first during the task. Aleatory variations in light intensity rather than progressive linear changes were initially discussed to control for possible habituation bias in participants. However, abrupt changes in light intensity and the reflected images could have been stressful for the participants. Moreover, in line with other studies [29, 75, 76], we presented the morphs incrementally from 0% self/other to 100% self/other in the two directions separately. This allowed us to differentiate the morphing directions and disentangle the two types of self/other confusion (i.e., egocentric bias and altercentric bias) and the critical thresholds for switching between self and other.

Similarly, the order of the presentation of the task was not randomized, which could have led to carry-over effects. However, there was no significant time effect on the results of the recognition task in the individuals with AN or the HCs, which allowed us to reduce the possible carryover and learning effects of the task.

## Conclusions

Although further research is needed to replicate these results, this study uncovered novel findings showing the first behavioural evidence of impaired SOD in individuals with AN through an embodied face-recognition paradigm generating a self–other face merging illusory effect in ecologically relevant conditions, i.e., when two individuals are physically facing each other and interacting.

This study is a first step in exploring the links between, on the one hand, body representation and sense of ownership, and, on the other hand, alterations in social interactions and cognitions that have unduly always been explored separately in the literature on AN.

Finally, the double mirror Alter Ego may also offer new therapeutic perspectives for AN based on a potential remediation pathway using double mirrors by practising control of representations of self and others and simultaneously improving the sense of body ownership, supporting self-differentiation, and correcting the poor self-perception that plays a causal role in the development, persistence, and relapse of AN.

## Supporting information

**S1 Raw data.**
(PDF)

## Acknowledgments

The authors thank the youth who participated in the study as well as Pierre Ailliot for the statistical analysis.

## Author Contributions

**Conceptualization:** Nathalie Lavenne-Collot, Emilie Maubant, Stéphanie Déroulez, Moritz Wehrmann, Alain Berthoz.

**Data curation:** Nathalie Lavenne-Collot, Emilie Maubant.

**Formal analysis:** Nathalie Lavenne-Collot, Emilie Maubant, Stéphanie Déroulez.

**Investigation:** Emilie Maubant.

**Methodology:** Nathalie Lavenne-Collot, Stéphanie Déroulez.

**Project administration:** Michel Botbol.

**Resources:** Alain Berthoz.

**Software:** Moritz Wehrmann.

**Supervision:** Michel Botbol, Alain Berthoz.

**Validation:** Nathalie Lavenne-Collot, Stéphanie Déroulez.

**Writing – original draft:** Nathalie Lavenne-Collot, Michel Botbol, Alain Berthoz.

**Writing – review & editing:** Nathalie Lavenne-Collot, Guillaume Bronsard, Michel Botbol, Alain Berthoz.

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
