## [Decision Letter · Decision Letter 0]

22 Nov 2023

PONE-D-23-07323Self /other recognition and distinction in adolescents with Anorexia Nervosa assessed with a double mirror paradigmPLOS ONE

Dear Dr. Lavenne-Collot,

Thank you for submitting your manuscript to PLOS ONE. After careful consideration, we feel that it has merit but does not fully meet PLOS ONE’s publication criteria as it currently stands. Therefore, we invite you to submit a revised version of the manuscript that addresses the points raised during the review process.

**Reviewers considered the topic innovative and relevant, although several concerns have been expressed concerning the quality of the methodology. An essential point suggested by the reviewers is the inclusion of new participants in order to get to a minimun of 15 participants per group, as well as enhancing the exploration of the topic with new assessment instruments.**

We look forward to receiving your revised manuscript.

Kind regards,

Inmaculada Riquelme

Academic Editor

PLOS ONE

Journal Requirements:

2. We noted in your submission details that a portion of your manuscript may have been presented or published elsewhere. "We have submitted to PlosOne an article using the same double mirror protocol entitled "Self/other distinction in adolescents with autism spectrum

disorder (ASD) assessed with a double mirror paradigm" which is currently in press and will be published shortly. Some of the figures or sentences in the text are therefore similar. However, the study population is not the same, and the paradigm has evolved by introducing new tasks and another focus in the case of anorexia compared to ASD." Please clarify whether this [conference proceeding or publication] was peer-reviewed and formally published. If this work was previously peer-reviewed and published, in the cover letter please provide the reason that this work does not constitute dual publication and should be included in the current manuscript.

Reviewers' comments:

Reviewer's Responses to Questions

**Comments to the Author**

1. Is the manuscript technically sound, and do the data support the conclusions?

Reviewer #1: Yes

Reviewer #2: Partly

2. Has the statistical analysis been performed appropriately and rigorously? 

Reviewer #1: I Don't Know

Reviewer #2: No

3. Have the authors made all data underlying the findings in their manuscript fully available?

Reviewer #1: Yes

Reviewer #2: No

4. Is the manuscript presented in an intelligible fashion and written in standard English?

Reviewer #1: Yes

Reviewer #2: Yes

5. Review Comments to the Author

Reviewer #1: The full review is also attached.

Review PLOS ONE manuscript (PONE-D-23-07323) July 2023

Summary. The aim of this pilot study was to examine self-recognition and self/other distinction (SOD) in seven adolescents with anorexia nervosa (AN) and in seven matching controls (HC) by using a self-versus other face identification task through a double mirror ”Alter Ego”TM device (developed by Moritz Wehrmann) that provide an ability “to progressively change the identity of self and other in the mirror with changes in lighting on both sides of the device” (p. 7).

The authors presents previous and relevant research on body image and body schema issues, and discuss limitations associated with dysfunctional beliefs and cognitions associated with body image, and lack of reference to social processes involved in the relations between self and others in research focusing on the representation of the physical body and body schema issues. The authors then offer a reasoning for addressing self/other distinction in individuals with AN. Self/other distinction and the ability to “differentiate one´s own body, actions, and mental states from those of others is crucial for establishing relationships with others while maintaining a stable sense of self” (p.4). In lacking such capacity, confusion between self and other may occur. Previous research on SOD in the general population and in patients with mental disorders (including AN) have mostly used static images or movies, and are considered insufficient for SOD assessment. It is argued that to better understand the underpinnings of AN, studies should take on an embodied perspective involving participants who are physically present.

Criteria for inclusion and exclusion of participants and controls are well described. Descriptions of the experiment set-up, and the two different procedures, the neutral condition and the sensorimotor condition, are sound and convincing. The two judgement criteria used were: M1, the threshold at which subjects start to recognize their own face during other-to-self morphing, and M2, the threshold at which subjects start to recognize the other´s face during self-to-other morphing. A specialist in statistics has assisted with the statistical analysis.

The results from neutral condition showed that earlier self-recognition in the other-to-self and delayed other-recognition in the self-to-other sequence in participants with AN compared to the HC. As opposed to that of controls, the critical threshold for switching between self and other varied with the direction of morphing in participants with AN.

The main finding measured during the sensorimotor condition, when the subjects with AN were seated in a chair where they were touching the back- and footrest (BFR) and receiving some postural support, the self-recognition threshold (M1) increased significantly and was nearer to that of controls. In this position, the AN subjects showed delayed self-recognition as they continued to perceive the other longer while their own image was gradually appearing. The findings and their implications are thoroughly discussed in relation to other research, and it is theorized whether the use of postural support might have affected self-recognition and SOD in subjects with AN.

The focus of the paper is recognition of self and other in individuals with AN. The results suggest that disturbances in mirror self-recognition indicates bodily self-consciousness impairments in subjects with AN, that subjects with AN are more egocentric in SOD compared to HC´s, and that self-recognition and SOD thresholds change when postural conditions in subjects with AN are modified. The theme of the paper is important and the study contributes to increased understanding of SOD in individuals with AN. It is well written and it refer to a number work relating to SOD, while some theoretical and conceptual aspects needs some further elaboration. Having done that, the article would be recommended for publication.

Concerns

There are two concerns related to the theoretical and conceptual foundations of the study:

1. What about the self in individuals with AN? In the presentation of the rationale behind exploring self/other distinction in individuals with AN, the authors claim that a stable sense of self is crucial in order to relate to the social world and establish relationships with others, as well as for differentiating one´s own body from those of others (p. 4). It could, however, also be argued that to be able to differentiate oneself from the other, there must be a stable sense of self. But what is it that constitutes a “stable sense of self” in this context? Having a psychiatric disorder involve transformations to one´s sense of self and one´s body, and the way one inhabits, or live, one´s body is changed individuals with AN. See for example Legrand (1), which you already have referenced, who claims that people with AN often lack the experience of the body as an integrated place of their own. How can these ideas affect the ability to accurately recognize others?

2. The second concern is about the concept of embodiment and whether “involving participants who are physically present” is equivalent to an embodied perspective.

The embodiment thesis infers that the subject (and perception of the self) is constitutively bodily (2, p 208). To be an experiencing subject, one needs to be an experiencing body as well. Even though the subjects were physically present in this study, were they also embodied, in the sense that the body is the source of subjectivity, feelings, perceptions and sensations? Given that only the participants head and faces were visual in the experiments, did this speak more to visual recognition and cognition than proprioception and embodiment? As the results of the study show, when providing the subjects with backrest and footrest, and the possibility of receiving bodily support from these, bodily tension (AN is strongly associated with high levels of anxiety, which tend to produce postural stiffness, muscular tension, and constricted breathing) might have lessened and contributed to the subjects being more present in the here and now. The authors of a recent article on the lived body, found that in allowing themselves to take support from the seat and back of the chair (as part of physiotherapy consultations, which also involved other means), instead of perching on the edge of the chair, subjects with AN felt they gained a feeling of becoming more attentive and of being more present in the here and now (3). Given that embodiment, perception, action and subjectivity are inseparable, such changes to the body may thus invoke changes in one´s sense of self and one´s way of being in the world.

Other points

This reviewer have limited competence in statistics, but have confidence in the work done by the specialist and that the statistics will be considered by other reviewers.

A few spelling issues:

p 11, line 13: strategies (lack r),

p 14, second line in section on “Neutral condition”: “individuals with AN” x 2?

References

1. Legrand D. Subjective and physical dimensions of bodily self-consciousness, and their dis-integration in anorexia nervosa. Neuropsychologia. 2010;48:726-37.

2. Legrand D. Phenomenological dimensions of bodily self-consciousness. In: Gallagher S, editor. The Oxford Handbook of the Self. New York: Oxford university press; 2011. p. 204-27.

3. Naess CR, Kolnes L-J. A preliminary exploration of experiences of integrating the body in the self in two women with anorexia nervosa in view of phenomenological conceptualisations. Journal of eating disorders. 2022;10.

Reviewer #2: I suggest to include other participants to impove the realibility and generalization of your results. This would allow to perform better statistical analysis. I have some concerns related to the second research question, where there is no a control group. Moreover i suggest to improve the literature search, since some statements might result too strong in light of recent evidence.

6. PLOS authors have the option to publish the peer review history of their article (what does this mean?). If published, this will include your full peer review and any attached files.

Reviewer #1: No

Reviewer #2: No

---

## [Author Response · Author response to Decision Letter 0]

8 Feb 2024

RESPONSE TO REVIEWERS

REVIEWER 1

Summary. The aim of this pilot study was to examine self-recognition and self/other distinction (SOD) in seven adolescents with anorexia nervosa (AN) and in seven matching controls (HC) by using a self-versus other face identification task through a double mirror ”Alter Ego”TM device (developed by Moritz Wehrmann) that provide an ability “to progressively change the identity of self and other in the mirror with changes in lighting on both sides of the device” (p. 7). 

The authors presents previous and relevant research on body image and body schema issues, and discuss limitations associated with dysfunctional beliefs and cognitions associated with body image, and lack of reference to social processes involved in the relations between self and others in research focusing on the representation of the physical body and body schema issues. The authors then offer a reasoning for addressing self/other distinction in individuals with AN. Self/other distinction and the ability to “differentiate one´s own body, actions, and mental states from those of others is crucial for establishing relationships with others while maintaining a stable sense of self” (p.4). In lacking such capacity, confusion between self and other may occur. Previous research on SOD in the general population and in patients with mental disorders (including AN) have mostly used static images or movies, and are considered insufficient for SOD assessment. It is argued that to better understand the underpinnings of AN, studies should take on an embodied perspective involving participants who are physically present. 

Criteria for inclusion and exclusion of participants and controls are well described. Descriptions of the experiment set-up, and the two different procedures, the neutral condition and the sensorimotor condition, are sound and convincing. The two judgement criteria used were: M1, the threshold at which subjects start to recognize their own face during other-to-self morphing, and M2, the threshold at which subjects start to recognize the other´s face during self-to-other morphing. A specialist in statistics has assisted with the statistical analysis. 

The results from neutral condition showed that earlier self-recognition in the other-to-self and delayed other-recognition in the self-to-other sequence in participants with AN compared to the HC. As opposed to that of controls, the critical threshold for switching between self and other varied with the direction of morphing in participants with AN. 

The main finding measured during the sensorimotor condition, when the subjects with AN were seated in a chair where they were touching the back- and footrest (BFR) and receiving some postural support, the self-recognition threshold (M1) increased significantly and was nearer to that of controls. In this position, the AN subjects showed delayed self-recognition as they continued to perceive the other longer while their own image was gradually appearing. The findings and their implications are thoroughly discussed in relation to other research, and it is theorized whether the use of postural support might have affected self-recognition and SOD in subjects with AN. 

The focus of the paper is recognition of self and other in individuals with AN. The results suggest that disturbances in mirror self-recognition indicates bodily self-consciousness impairments in subjects with AN, that subjects with AN are more egocentric in SOD compared to HC´s, and that self-recognition and SOD thresholds change when postural conditions in subjects with AN are modified. The theme of the paper is important and the study contributes to increased understanding of SOD in individuals with AN. It is well written and it refer to a number work relating to SOD, while some theoretical and conceptual aspects needs some further elaboration. Having done that, the article would be recommended for publication.

Concerns

There are two concerns related to the theoretical and conceptual foundations of the study:

1. What about the self in individuals with AN? In the presentation of the rationale behind exploring self/other distinction in individuals with AN, the authors claim that a stable sense of self is crucial in order to relate to the social world and establish relationships with others, as well as for differentiating one´s own body from those of others (p. 4). It could, however, also be argued that to be able to differentiate oneself from the other, there must be a stable sense of self. But what is it that constitutes a “stable sense of self” in this context? Having a psychiatric disorder involve transformations to one´s sense of self and one´s body, and the way one inhabits, or live, one´s body is changed individuals with AN. See for example Legrand (1), which you already have referenced, who claims that people with AN often lack the experience of the body as an integrated place of their own. How can these ideas affect the ability to accurately recognize others? 

First of all, the authors would like to thank the reviewer for his very attentive proofreading and pertinent comments, particularly with regard to his knowledge of the phenomenological approach to anorexia, which fits in perfectly with our approach using the double mirror in this disorder.

Indeed, certain theoretical and conceptual aspects deserved to be developed, in particular by emphasizing the phenomenological approach to anorexia, which allows access to an integrative and multidimensional conception of bodily self-consciousness that does not leave out the interpersonal and intersubjective aspects of the disorder.

This has enabled us to address the problem of objectification of the body and the tension between the first-person lived body and the third-person object body, as well as to introduce more clearly the question of the relevance of anomalies in the boundaries between self/other in anorexia nervosa within the introduction section. 

2. The second concern is about the concept of embodiment and whether “involving participants who are physically present” is equivalent to an embodied perspective. 

The embodiment thesis infers that the subject (and perception of the self) is constitutively bodily (2, p 208). To be an experiencing subject, one needs to be an experiencing body as well. Even though the subjects were physically present in this study, were they also embodied, in the sense that the body is the source of subjectivity, feelings, perceptions and sensations? Given that only the participants head and faces were visual in the experiments, did this speak more to visual recognition and cognition than proprioception and embodiment? 

Given that only the participants' heads and faces were visualized in our experiment, the author of the article raises the question of whether our study is more about visual recognition and cognition than proprioception and embodiment. 

Firstly, to take account of this pertinent comment, we have added a paragraph highlighting the special status of recognizing one's own face in the mirror from a phenomenological perspective as follows: 

« From a phenomenological perspective, the image reflected in the mirror also reveals the ambiguity of realizing one's body as an object visible from the outside or by others, and yet being that object oneself - in other words, a tension between a primary embodied first person perspective and an internalized third-person perspective (Mancini, 2021). This intrapersonal tension also becomes an interpersonal one : while realizing that I am objectified or even reified by the gaze of the other, I must resist this gaze and (re)assert my own subjectivity, even if it is by objectifying the other in turn (Sartre, 2018). »

Furthermore, from a neuroscientific perspective, several studies have also provided important insights into how face identification is linked not only to visual or cognitive skills, but also to the multisensory integration of body-related signals in healthy volunteers (Tajadura-Jiménez et al., 2012). Indeed, it has been shown that face identification integrates not only visual information, but also kinesthetic, proprioceptive, visceroceptive and tactile, visual and tactile data, similar to bodily self-consciousness (Tsakiris, 2008; Sforza et al., 2009; Paladino et al., 2010; Apps et al..., 2015). Furthermore, using the double-mirror paradigm, Thirioux et al. demonstrated bidirectional links between face recognition in the mirror and bodily self-consciousness by showing that changes in self-face identification induce a bottom-up shift between the current visual representation of one's own face and the memorized representation of one's own face, which in turn has a top-down impact on bodily self-consciousness.

As the results of the study show, when providing the subjects with backrest and footrest, and the possibility of receiving bodily support from these, bodily tension (AN is strongly associated with high levels of anxiety, which tend to produce postural stiffness, muscular tension, and constricted breathing) might have lessened and contributed to the subjects being more present in the here and now. The authors of a recent article on the lived body, found that in allowing themselves to take support from the seat and back of the chair (as part of physiotherapy consultations, which also involved other means), instead of perching on the edge of the chair, subjects with AN felt they gained a feeling of becoming more attentive and of being more present in the here and now (3). Given that embodiment, perception, action and subjectivity are inseparable, such changes to the body may thus invoke changes in one´s sense of self and one´s way of being in the world. 

A small number of studies exist in which the aim was to enhance body awareness (often referred to as ‘body awareness therapies’) and to apply body-oriented interventions for subjects with AN.

The authors would like to thank the reviewer for his in-depth knowledge of the literature, and this bibliographic recommendation, which provides useful support for their findings. The results of this study have been added to the discussion section.

Other points

This reviewer have limited competence in statistics, but have confidence in the work done by the specialist and that the statistics will be considered by other reviewers. 

The authors thank the reviewer for his confidence

A few spelling issues:

p 11, line 13: strategies (lack r), 

p 14, second line in section on “Neutral condition”: “individuals with AN” x 2?

The authors thank the reviewer for his vigilance. Corrections have been made to the manuscript

References

1. Legrand D. Subjective and physical dimensions of bodily self-consciousness, and their dis-integration in anorexia nervosa. Neuropsychologia. 2010;48:726-37.

2. Legrand D. Phenomenological dimensions of bodily self-consciousness. In: Gallagher S, editor. The Oxford Handbook of the Self. New York: Oxford university press; 2011. p. 204-27.

3. Naess CR, Kolnes L-J. A preliminary exploration of experiences of integrating the body in the self in two women with anorexia nervosa in view of phenomenological conceptualisations. Journal of eating disorders. 2022;10.

The authors would like to thank the reviewer for these suggestions, which helped to complete their bibliography.

Reviewer 2

Thank you for giving me the opportunity to review this interesting study. In my opinion, the study is very innovative and original. However, I have some concerns about the current manuscript.

The authors would like to thank the reviewer most sincerely for his careful proofreading and his well-informed comments on a very specific and original topic.

Abstract

• The abstract could be improved 

• Anorexia Nervosa should be written with upper cases

• I suggest to use the expression “Participants affected by AN” instead of “with AN”

The authors would like to thank the reviewer for his comments:

The abstract has been improved: Anorexia nervosa was written with upper cases and the expression "Participants affected by anorexia nervosa" has replaced "with AN".

Introduction

• It is not clear the link between body distortion and self-others impairments (pag. 2/3) that lead authors to their research question. Maybe this relationship might be better explained to the reader.

The authors would like to thank the reviewer for this request for clarification regarding the relationship between body distortion abnormalities in anorexia nervosa and self-others impairments. 

In particular, this relationship is based on an abundant literature that emphasizes the central role of the body in establishing a clear distinction between self and other (Paladino et al., 2010, Tajadura-Jiménez A et al. 2012, Tsakiris et al., 2017, Palmer &Tsakiris, 2018).Especially, it is bodily self-consciousness based on multisensory integration processes that underpins the distinction between self and other (Keromnes et al., 2019). The presentation of multisensory integration anomalies previously described in AN therefore raises the question of possible anomalies in the distinction between self and other.

This transition between body distortion abnormalities and SOD impairments has been made more explicit for the reader by the addition of an extra paragraph within the introduction

Methods

• The sample size is very small. We suggest to recruit participants, in order to have at least 15 participants for group. Because of the interesting topic and the novelty of the methology, this would improve the value of this manuscript.

The authors thank the reviewer for this comment. We are well aware of the small sample size. However, for logistical reasons, it was impossible to include more participants and we are currently unable to complete recruitment to meet the recommended target. 

Nevertheless, several arguments can be advanced to support the publication of this work despite the small sample size: 

-Firstly, Anderson and Vingrys have argued that "small samples may be sufficient to show the presence of an effect, but not to estimate the size of the effect. If the objective is only to show the existence of an effect, it is possible to avoid the cost of a large sample." In our study, the effect is quite marked, and in any case sufficient to obtain p-values <0.05. 

This argument as well as bibliographic references has been added to the manuscript

Anderson AJ, Vingrys AJ. Small samples: does size matter? Invest Ophthalmol Vis Sci. 2001 Jun;42(7):1411-3. PMID: 11381039.

Indrayan A, Mishra A. The importance of small samples in medical research. J Postgrad Med. 2021 Oct-Dec;67(4):219-223. doi: 10.4103/jpgm.JPGM_230_21. PMID: 34845889; PMCID: PMC8706541.

- Secondly, we stressed several times in the manuscript that this was a pilot study, and we moderated the value of our results due to the small sample size.

-Third, a similar pilot study involving exactly the same small number of subjects with Autism Spectrum Disorders was accepted and published in the journal PlosOne (Lavenne, 2022) with a similar methodology, emphasizing that the small number of participants was not a sufficient obstacle to oppose publication.

• Moreover, no pathological features are reported (subtype of diagnosis, current pharmacological treatment).

We thank the reviewer for this comment. We have added data concerning the pathological characteristics of participants with AN (table 1) and specified that the participants were not taking any medication.

• I suggest to include other measures based on literature: for instance, body distortion might be associated with altered SOD. 

The authors thank the reviewer for this comment. It would indeed be very interesting to include other measures based on the literature, for example concerning body distortion, which could be associated with alterations in SOD. However, this goes far beyond the scope of this preliminary work, which was to provide the first evidence of SOD alterations in anorexia nervosa and to characterize their type (egocentrism / heterocentrism). This proposal by the reviewer may, however, be the subject of future work.

• I would evaluate also visual abilities across groups (use of lens, glasses, other visual-corrective devices)

---

## [Decision Letter · Decision Letter 1]

18 Apr 2024

PONE-D-23-07323R1Self /other recognition and distinction in adolescents with Anorexia Nervosa assessed with a double mirror paradigmPLOS ONE

Dear Dr. Lavenne-Collot,

Thank you for submitting your manuscript to PLOS ONE. After careful consideration, we feel that it has merit but does not fully meet PLOS ONE’s publication criteria as it currently stands. Therefore, we invite you to submit a revised version of the manuscript that addresses the points raised during the review process.

Some circunstances during the review process produced a delay in the revision. Reviewer 2 was not available and the revised version of your manuscript was sent to another reviewer. Reviewer 3 is very positive regarding your innovative study, but has some suggestions for further improving the quality of your manuscript.

We look forward to receiving your revised manuscript.

Kind regards,

Inmaculada Riquelme

Academic Editor

PLOS ONE

Reviewers' comments:

Reviewer's Responses to Questions

**Comments to the Author**

1. If the authors have adequately addressed your comments raised in a previous round of review and you feel that this manuscript is now acceptable for publication, you may indicate that here to bypass the “Comments to the Author” section, enter your conflict of interest statement in the “Confidential to Editor” section, and submit your "Accept" recommendation.

Reviewer #1: All comments have been addressed

Reviewer #3: (No Response)

2. Is the manuscript technically sound, and do the data support the conclusions?

Reviewer #1: Yes

Reviewer #3: Partly

3. Has the statistical analysis been performed appropriately and rigorously? 

Reviewer #1: I Don't Know

Reviewer #3: No

4. Have the authors made all data underlying the findings in their manuscript fully available?

Reviewer #1: Yes

Reviewer #3: No

5. Is the manuscript presented in an intelligible fashion and written in standard English?

Reviewer #1: Yes

Reviewer #3: Yes

6. Review Comments to the Author

Reviewer #1: The authors have addressed all items that were identified as issues in the first iteration of the draft. I think this piece is ready for publication.

Reviewer #3: I really like the idea behind this paper, the innovative approach, and the findings are of interest for a broader audience.

I have some suggestions to improve the work and some questions.

- Please add "pilot study" to the title and include the n in the abstract

- You indicate that all the data is available, but this does not seem to be the case. I do not find any raw data in the supplement.

- You give a very nice overview of the existing literature, except you are missing a whole large field of studies that are very important with regard to body perception, self-other-distinction, and sensorimotor integration. There are many studies on the perception and processing of social, affective touch, which has been suggested to play a crucial role in the development of the bodily self and might specifically altered in anorexia. These findings should be integrated bopth in the introduction and in the discussion

Here are some references:

Crucianelli, L., Cardi, V., Treasure, J., Jenkinson, P. M., & Fotopoulou, A. (2016). The perception of affective touch in anorexia nervosa. Psychiatry research, 239, 72-78.

Frost-Karlsson, M., Capusan, A. J., Perini, I., Olausson, H., Zetterqvist, M., Gustafsson, P. A., & Boehme, R. (2022). Neural processing of self-touch and other-touch in anorexia nervosa and autism spectrum condition. NeuroImage: Clinical, 36, 103264.

Davidovic, M., Karjalainen, L., Starck, G., Wentz, E., Björnsdotter, M., & Olausson, H. (2018). Abnormal brain processing of gentle touch in anorexia nervosa. Psychiatry Research: Neuroimaging, 281, 53-60.

Bischoff-Grethe, A., Wierenga, C. E., Berner, L. A., Simmons, A. N., Bailer, U., Paulus, M. P., & Kaye, W. H. (2018). Neural hypersensitivity to pleasant touch in women remitted from anorexia nervosa. Translational psychiatry, 8(1), 161.

Bellard, A., Trotter, P., McGlone, F., & Cazzato, V. (2022). Vicarious ratings of self vs. other-directed social touch in women with and recovered from Anorexia Nervosa. Scientific Reports, 12(1), 13429.

Boehme, R., & Olausson, H. (2022). Differentiating self-touch from social touch. Current Opinion in Behavioral Sciences, 43, 27-33.

Ciaunica, A., Constant, A., Preissl, H., & Fotopoulou, K. (2021). The first prior: from co-embodiment to co-homeostasis in early life. Consciousness and cognition, 91, 103117.

- Do you have the BMI values for the controls?

- Did you colllect Autism quotient scores?

- I find the description of the tasks somewhat confusing, i.e. it is unclear how often the back and forth passage occurs. Maybe you could start with a simple overview over the procedure (how often is each condition repeated). This is also unclear for the sensorimotor stimulations. Were they all just presented once? Is it always a back and forth neutral followed by one of the sensorimotor conditions?

- How long did each trial last? How long was the complete experiment?

- Were participants allowed to move / grimace during the tasK? What was the instruction?

- Do I see this correctly, that you do not correct for multiple comparisons?

- An alternative explanantion could be that the AN participants are worse at matching their actions with the sensory input, i.e. they have problems with predicting the sensory outcome of their own action (so they do not identify when the mirror image does not match their movement - even if they kept their face still, there is still movement due to blinking the eyes, breathing etc).

- Another alternative that I do not see can be controlled for is: I assume anorexia patients do not like seeing their own face. This could mean a) they do not often look in the mirro and are simply not used to seeing tehir won face (however, then they would probably tend to identify the other faster), or b) they really dislike seeing their own face and are therefore more sensitive to perceiving it.

- The sensorimotor feedback from the back and foot support could also simply be a distraction from the over-focus on the own face?

minor:

- introduction: third line, should be "a growing number of studies"

- I am also not a native English speaker, so pardon me if this is incorrect, but I find the formulation "AN individuals were installed in the BFR condition" very strange.

7. PLOS authors have the option to publish the peer review history of their article (what does this mean?). If published, this will include your full peer review and any attached files.

Reviewer #1: No

Reviewer #3: No

---

## [Author Response · Author response to Decision Letter 1]

22 Jul 2024

Reviewer #1: The authors have addressed all items that were identified as issues in the first iteration of the draft. I think this piece is ready for publication.

The authors would like to thank the reviewer for his comments and advice on the first draft, which helped to improve the article. They are also very grateful for his support and interest in our work.

Reviewer #3: I really like the idea behind this paper, the innovative approach, and the findings are of interest for a broader audience.

I have some suggestions to improve the work and some questions.

First of all, the authors would like to thank the reviewer for his positive comments, for the interest and time he devoted to their work and, above all, for his in-depth knowledge of the issue, which enabled them to significantly improve the article. They hope that the response to his comments and the changes they have made will confirm that this work is worthy of publication.

- Please add "pilot study" to the title and include the n in the abstract

The authors would like to thank the reviewer for this comment. The qualifier "pilot" has been added to the title and the n included in the abstract.

- You indicate that all the data is available, but this does not seem to be the case. I do not find any raw data in the supplement.

The authors would like to thank the reviewer for this comment. Raw data have been added as a supplement

- You give a very nice overview of the existing literature, except you are missing a whole large field of studies that are very important with regard to body perception, self-other-distinction, and sensorimotor integration. There are many studies on the perception and processing of social, affective touch, which has been suggested to play a crucial role in the development of the bodily self and might specifically altered in anorexia. These findings should be integrated both in the introduction and in the discussion

Here are some references:

Crucianelli, L., Cardi, V., Treasure, J., Jenkinson, P. M., & Fotopoulou, A. (2016). The perception of affective touch in anorexia nervosa. Psychiatry research, 239, 72-78.Frost-Karlsson, M., Capusan, A. J., Perini, I., Olausson, H., Zetterqvist, M., Gustafsson, P. A., & Boehme, R. (2022). Neural processing of self-touch and other-touch in anorexia nervosa and autism spectrum condition. NeuroImage: Clinical, 36, 103264.

Davidovic, M., Karjalainen, L., Starck, G., Wentz, E., Björnsdotter, M., & Olausson, H. (2018). Abnormal brain processing of gentle touch in anorexia nervosa. Psychiatry Research: Neuroimaging, 281, 53-60.

Bischoff-Grethe, A., Wierenga, C. E., Berner, L. A., Simmons, A. N., Bailer, U., Paulus, M. P., & Kaye, W. H. (2018). Neural hypersensitivity to pleasant touch in women remitted from anorexia nervosa. Translational psychiatry, 8(1), 161.

The authors would like to thank the reviewer for this very pertinent commentary and additional bibliography. We have integrated the literature about alterations in social and affective touch in anorexia, emphasizing its crucial role in the development of the bodily self and in the self/other distinction, both in the introduction and in the discussion with two additional paragraphs added on pages 6 and 23 respectively. All the suggested bibliography, and even more, has been added.

- Do you have the BMI values for the controls?

BMI values for the controls were not analyzed. However, it was checked at inclusion that controls had a normal BMI in addition to the absence of eating disorders. This was specified in the methodology section (page 9).

- Did you colllect Autism quotient scores?

We thank the reviewer for this question. We did not collect participants' autistic quotient scores, but the reviewer is right to point out that this would have been relevant. 

Indeed, there is an overlap in symptomatology between ASD and anorexia nervosa. Moreover, in a previous study using the Alter ego double mirror system, we showed alterations in the self/other distinction in participants with ASD compared with controls.

 However, the aim of this work was above all to demonstrate for the first time the existence of SOD in AN. Consequently, the question of comparing these two populations was beyond the scope of this pilot study, but deserves to be explored in a future study. This has been added to the manuscript (page 26).

- I find the description of the tasks somewhat confusing, i.e. it is unclear how often the back and forth passage occurs. Maybe you could start with a simple overview over the procedure (how often is each condition repeated). This is also unclear for the sensorimotor stimulations. Were they all just presented once? Is it always a back and forth neutral followed by one of the sensorimotor conditions?

We thank the reviewer for this comment. 

We have improved the description of the procedure by starting with a simple overview at the beginning of the "protocol and task" section (page 11) as below:

The experimental procedure had a duration of approximately 90 minutes and was divided into 2 parts:

- First, the neutral condition, lasted 15 minutes and consisted of a « back-and-forth » passage, i.e. TDCs moved progressively from self to other, then from other to self while, during the same period of time, participants with AN experienced the opposite condition, i.e. they moved from other to self, then from self to other ;

-The second part, called the sensorimotor stimulation condition, lasted 60 minutes and involved exposing participants to the same light intensity conditions as described in the neutral condition, i.e. a "back-and-forth » passage, but repeated 5 times with a different stimulus applied to the participants with AN during each passage.

Thus, the whole task consisted of 6 back-and-forth passages: one in neutral condition, then 5 with different successive sensory stimuli only for AN participants. 

There was a 10-min pause between the first and second parts to allow attentional recovery. 

- How long did each trial last? How long was the complete experiment?

We thank the reviewer for this comment. The duration of each test has been specified in the manuscript as indicated in the new paragraph above (Page 11).

- Were participants allowed to move / grimace during the tasK? What was the instruction?

We thank the reviewer for this comment. It was specified in the methodology section that participants were positioned so as to allow strict alignment of the eyes of both partners in the mirror and that they were asked to maintain this alignment throughout the task, without making any body or facial movements (including no grimace). Moreover, the instruction given to participants was clarified (page 11).

- Do I see this correctly, that you do not correct for multiple comparisons? 

The reviewer is right, we made no multiple comparisons adjustments. Indeed, adjustments for making multiple comparisons in large bodies of data are recommended to avoid rejecting the null hypothesis too readily. Howewer, a policy of not making adjustments for multiple comparisons is sometimes preferable because it will lead to fewer errors of interpretation when the data under evaluation are not random numbers but actual observations on nature. (Rothman, 1990, Barnett et al., 2022).

Rothman, K.J. (1990). No adjustments are needed for multiple comparisons .Epidemiology, 1: 43-46.

Barnett, M. J., Doroudgar, S., Khosraviani, V., & Ip, E. J. (2022). Multiple comparisons: To compare or not to compare, that is the question. Research in social & administrative pharmacy : RSAP, 18(2), 2331–2334. https://doi.org/10.1016/j.sapharm.2021.07.006

- An alternative explanation could be that the AN participants are worse at matching their actions with the sensory input, i.e. they have problems with predicting the sensory outcome of their own action (so they do not identify when the mirror image does not match their movement - even if they kept their face still, there is still movement due to blinking the eyes, breathing etc).

The authors thank the reviewer for this comment. This additional hypothesis has been added to the discussion (page 23).

- Another alternative that I do not see can be controlled for is: I assume anorexia patients do not like seeing their own face. This could mean a) they do not often look in the mirror and are simply not used to seeing their own face (however, then they would probably tend to identify the other faster), or b) they really dislike seeing their own face and are therefore more sensitive to perceiving it.

We thank the reviewer for this comment. We believe that our results can not be explained by an attentional or emotional bias related to the fact that subjects with AN dislike seing their own face in the mirror (i.e. they would have avoided or be more sensitive to seeing it ). Indeed, if this hypothesis were correct, this tendency should have been the same regardless of the direction of morphing.

The answer to this question is now included in the "limitation" section of our results, when we state that our results cannot be explained by differences in attentional performance or information processing (page 27).

- The sensorimotor feedback from the back and foot support could also simply be a distraction from the over-focus on the own face?

We thank the author for this comment.

We believe that the answer we gave to the previous question also applies here: if the hypothesis was that of distraction due to excessive focus on one's own face, this effect should have been found similarly regardless of the direction of morphing.

Moreover, if the distraction hypothesis raised by the reviewer were correct, we should have expected an even greater distraction effect for the most unusual sensorimotor tasks (such as sitting on an unstable cushion). However, it is the effect of the task reinforcing the median axis of the body (which is not the most unusual one) that stands out significantly in our results, and confirms the clinical hypothesis of body therapists specialized in anorexia, even if our results still need to be replicated on a larger sample.

This has been added to the manuscript (page 28).

Minor:

- introduction: third line, should be "a growing number of studies"

The authors thank the reviewer for this remark. The wording has been improved.

- I am also not a native English speaker, so pardon me if this is incorrect, but I find the formulation "AN individuals were installed in the BFR condition" very strange.

The authors thank the reviewer for this remark. The wording has been improved (pages 17, 18 and 23).

---

## [Decision Letter · Decision Letter 2]

14 Aug 2024

Self /other recognition and distinction in adolescents with Anorexia Nervosa: A pilot study using a double mirror paradigm

PONE-D-23-07323R2

Dear Dr. Lavenne-Collot,

We’re pleased to inform you that your manuscript has been judged scientifically suitable for publication and will be formally accepted for publication once it meets all outstanding technical requirements.

Kind regards,

Inmaculada Riquelme

Academic Editor

PLOS ONE

Additional Editor Comments (optional):

Reviewers' comments:

Reviewer's Responses to Questions

**Comments to the Author**

1. If the authors have adequately addressed your comments raised in a previous round of review and you feel that this manuscript is now acceptable for publication, you may indicate that here to bypass the “Comments to the Author” section, enter your conflict of interest statement in the “Confidential to Editor” section, and submit your "Accept" recommendation.

Reviewer #3: All comments have been addressed

2. Is the manuscript technically sound, and do the data support the conclusions?

Reviewer #3: Yes

3. Has the statistical analysis been performed appropriately and rigorously? 

Reviewer #3: Yes

4. Have the authors made all data underlying the findings in their manuscript fully available?

Reviewer #3: Yes

5. Is the manuscript presented in an intelligible fashion and written in standard English?

Reviewer #3: Yes

6. Review Comments to the Author

Reviewer #3: (No Response)

7. PLOS authors have the option to publish the peer review history of their article (what does this mean?). If published, this will include your full peer review and any attached files.

Reviewer #3: No

---

## [Editor Report · Acceptance letter]

26 Aug 2024

PONE-D-23-07323R2 

PLOS ONE

Dear Dr. Lavenne-Collot, 

I'm pleased to inform you that your manuscript has been deemed suitable for publication in PLOS ONE. Congratulations! Your manuscript is now being handed over to our production team.

Kind regards, 

on behalf of

Dr. Inmaculada Riquelme 

Academic Editor

PLOS ONE